# Protein visualization and manipulation in *Drosophila* through the use of epitope tags recognized by nanobodies

Jun Xu[1]*[†], Ah-Ram Kim[1†], Ross W Cheloha[2], Fabian A Fischer[2], Joshua Shing Shun Li[1], Yuan Feng[1], Emily Stoneburner[1], Richard Binari[1], Stephanie E Mohr[1,3], Jonathan Zirin[1,3], Hidde L Ploegh[2], Norbert Perrimon[1,3,4]*

[1]Department of Genetics, Harvard Medical School, Boston, United States; [2]Boston Children's Hospital and Harvard Medical School, Boston, United States; [3]Drosophila RNAi Screening Center, Harvard Medical School, Boston, United States; [4]Howard Hughes Medical Institute, Boston, United States

*For correspondence:
Jun_Xu@hms.harvard.edu (JX);
perrimon@genetics.med.
harvard.edu (NP)

[†]These authors contributed
equally to this work

**Competing interest:** The authors
declare that no competing
interests exist.

**Reviewing Editor:** K
VijayRaghavan, National
Centre for Biological Sciences,
Tata Institute of Fundamental
Research, India

**Abstract** Expansion of the available repertoire of reagents for visualization and manipulation of proteins will help understand their function. Short epitope tags linked to proteins of interest and recognized by existing binders such as nanobodies facilitate protein studies by obviating the need to isolate new antibodies directed against them. Nanobodies have several advantages over conventional antibodies, as they can be expressed and used as tools for visualization and manipulation of proteins in vivo. Here, we characterize two short (<15aa) NanoTag epitopes, 127D01 and VHH05, and their corresponding high-affinity nanobodies. We demonstrate their use in *Drosophila* for in vivo protein detection and re-localization, direct and indirect immunofluorescence, immunoblotting, and immunoprecipitation. We further show that CRISPR-mediated gene targeting provides a straightforward approach to tagging endogenous proteins with the NanoTags. Single copies of the NanoTags, regardless of their location, suffice for detection. This versatile and validated toolbox of tags and nanobodies will serve as a resource for a wide array of applications, including functional studies in *Drosophila* and beyond.

## Introduction

Conventional antibodies typically have a MW ~150–160 kDa and are composed of four polypeptides, two identical heavy chains and two identical light chains. The size and composition of conventional immunoglobulins impose limitations on their application to in vivo studies. The recent development of smaller and single-polypeptide recombinant protein binders, such as single-chain variable fragments (~25 kDa), single-domain antibodies or 'nanobodies' (~12–15 kDa), and designed ankyrin repeat proteins (18 kDa for five repeats), has enabled many new applications (*Harmansa and Affolter, 2018*). These new types of recombinant binders are small and stable molecules that can be encoded in the genomes of model organisms or cells. Moreover, the coding sequences of these binders can be fused to various effector domains, making them useful as tools for imaging and for regulating the function of target proteins of interest (POIs) in vivo (*Helma et al., 2015*; *Harmansa and Affolter, 2018*; *Aguilar et al., 2019*). For example, a protein binder fused to a fluorescent protein can be expressed in vivo, where it can then bind to an endogenous target protein, an epitope-tagged protein, or even a post-translational modification, thus allowing visualization of subcellular localization of the target (*Harmansa and Affolter, 2018*; *Aguilar et al., 2019*). This is not usually possible when using conventional antibodies, which fail to assemble in the reducing environment of the cytosol.

Among available protein binders, camelid-derived nanobodies are particularly useful, as they consist of a single monomeric variable antibody domain that is the product of selection in vivo. Nanobodies are no less specific than conventional antibodies. Given their small size, nanobodies are easy to express in *Escherichia coli*, either alone or fused to a fluorescent marker or enzyme. The small size of nanobodies also allows better super-resolution microscopy than antibody-based imaging (*Fornasiero and Opazo, 2015*; *Mikhaylova et al., 2015*; *Virant et al., 2018*; *Fang et al., 2018*), and enables binding to epitopes not accessible to full-length conventional antibodies. Because nanobodies are usually stable in the reducing environment of intracellular space and are encoded as a single polypeptide, nanobodies or nanobody fusion proteins can be expressed in eukaryotes and used for a number of in vivo applications (*Helma et al., 2015*).

Nanobodies are powerful tools for manipulation of protein function and localization, as has been illustrated using nanobodies against GFP. For example, GFP-tagged proteins can be degraded using a GFP-targeting nanobody fused to an E3 ligase component, an approach that has been used for studies in *Drosophila melanogaster*, *Caenorhabditis elegans*, and *Danio rerio* (*Caussinus et al., 2011*; *Wang et al., 2017*; *Yamaguchi et al., 2019*). GFP-tagged proteins can be re-localized using a GFP-targeting nanobody fused to sequences or domains that specify a particular subcellular localization (*Harmansa et al., 2015*; *Harmansa et al., 2017*). Many proteins in model organisms such as *Drosophila* have been tagged with GFP, suggesting general applicability of the approach. However, the fusion of a target protein with GFP is not necessarily compatible with all applications, a significant limitation of a GFP-targeted approach. GFP is a bulky (27 kDa) substituent that might affect function or localization of the tagged protein. In addition, maturation of the GFP chromophore is slow, limiting its use for the imaging of nascent proteins.

An alternative approach would be to combine conventional epitope tags with the advantages of nanobody-based targeting. Because of their small size, epitope tags are less likely than GFP to interfere with the overall structure of the tagged protein. Several nanobodies that recognize small epitope tags have been isolated, including tags known as BC2-tag, EPEA-tag, MoonTag, and ALFA-tag (*Traenkle et al., 2015*; *De Genst et al., 2010*; *Boersma et al., 2019*; *Tanenbaum et al., 2014*; *Götzke et al., 2019*; *Cheloha et al., 2020*). Some of these have been used to visualize and manipulate tagged proteins by changing their abundance or localization using tag-targeting nanobodies in mammalian cells (*Zhao et al., 2019*; *Vigano et al., 2021*). The recently developed ALFA nanobody (NbALFA) recognizes a short peptide of 13 amino acids and provides a system useful for immunoblotting, protein purification, and imaging (fixed or live cells) (*Götzke et al., 2019*). In addition, it has been recently reported that HA frankenbody, a single-chain variable fragment engineered from anti-HA antibody, can be used for live imaging and protein degradation in vivo in *Drosophila* (*Vigano et al., 2021*). However, to the best of our knowledge, short linear epitope tag-specific nanobodies have yet to be applied for use in vivo in *Drosophila* or other multicellular organisms.

To expand the repertoire of nanobody-recognized tags (NanoTags) and corresponding nanobody tools, we characterized two NanoTags, VHH05- and 127D01-tags, and their corresponding nanobodies, NbVHH05 (*Ling et al., 2019*) and Nb127D01 (*Bradley et al., 2015*), for cellular and in vivo studies in *Drosophila*. Both nanobodies can be genetically encoded as fluorescent protein fusions (i.e. chromobodies [CBs]) that enable the detection of target proteins that carry NanoTags at N-terminal, internal or C-terminal sites. We show that these nanobodies are useful for multiplexed immunostaining, immunoblotting, and immunoprecipitation. We show that NanoTagged proteins can be manipulated by nanobodies fused to various subcellular localization signals. Moreover, in transgenic flies that overexpress NanoTagged proteins, we confirmed that the tagged proteins can be detected using GFP-tagged nanobodies expressed in vivo (i.e. with CBs) or by immunostaining. Finally, using CRISPR-based genome engineering, we generated flies with NanoTags inserted into endogenous genes. Our data show that VHH05- and 127D01-tags and their corresponding nanobodies can be used effectively for labeling and manipulating proteins, providing powerful tools for functional studies in *Drosophila* and other organisms.

## Results

### Characterization of VHH05 and 127D01 NanoTags

We selected two nanobody/NanoTag pairs to test in *Drosophila*, NbVHH05/VHH05-tag and Nb127D01/127D01-tag. NbVHH05 is a 111 amino acid (aa) nanobody that recognizes a 14 aa sequence (VHH05-tag, QADQEAKELARQIS) derived from the human E2 ubiquitin-conjugating enzyme UBC6e (*Figure 1A and B*), and has a binding constant ($K_d$) for the VHH05-tag of ~0.15 nM (*Ling et al., 2019*). Nb127D01 is a 115 aa nanobody that binds to an extracellular portion of the human C-X-C chemokine receptor type 2 (CXCR2) (*Bradley et al., 2015*). As the extracellular region of CXCR2 is large, we reduced the epitope-binding region to a minimal sequence of 10 aa (127D01-tag, SFEDFWKGED) (*Figure 1C and D*; *Figure 1—figure supplement 1*) to obtain a more versatile tag, and determined the $K_d$ of Nb127D01 to this tag to be <50 nM (data not shown). Protein-protein BLAST searches with both NanoTags failed to identify fully homologous sequences in the *Drosophila* proteome, thus reducing the possibility of spurious cross-reactions with endogenous, untagged proteins.

To determine whether these NanoTags can be used for live imaging, we visualized both the Nano-Tags and their corresponding nanobodies concurrently. To do this, we constructed vectors that use the *Actin5c* promoter to ubiquitously express mCherry proteins equipped with the NanoTags and with different cell compartment localization sequences at the N-terminus: a cell membrane localization protein (murine CD8 gene, NM_009857.1), a mitochondrial outer membrane sequence (TM domain of *Homo sapiens* CDGSH iron sulfur domain 1, NM_018464.5), and a nuclear localization sequence (histone H2B gene, NM_001032214.2). We also constructed vectors that ubiquitously express GFP-tagged nanobodies (NbVHH05-GFP and Nb127D01-GFP) (*Figure 1—figure supplement 2A, B*) under the control of the *Actin5c* promoter.

When expressed alone in *Drosophila* S2*R*+ cells, NanoTagged mCherry fusion proteins were observed in the expected subcellular compartments *Figure 1—figure supplement 2C, D*), indicating that the VHH05- and 127D01-tags do not affect protein localization. When either NbVHH05-GFP or Nb127D01-GFP was expressed in cells, we observed a GFP signal in the nucleus and the cytoplasm, although some S2*R*+ cells transfected with Nb127D01-GFP contained aggregates (*Figure 1—figure supplement 2C, D*). Next, we co-transfected S2*R*+ cells with either mCherry-VHH05 fusion proteins and NbVHH05-GFP or mCherry-127D01 fusion proteins and Nb127D01-GFP. In all cases, the GFP and mCherry signals co-localized - with a distribution indistinguishable from that of mCherry fusions alone (*Figure 1—figure supplement 2E, F*). We also tested whether mCherry-tagged nanobodies (NbVHH05-mCherry or Nb127D01-mCherry) co-localized with mitochondrial GFP tagged with the VHH05-tag or the 127D01-tag, respectively. As expected, we observed co-localization of NbVHH05-mCherry and mito-GFP-VHH05 when expression vectors for each were co-transfected in S2*R*+ cells (*Figure 1—figure supplement 2G*. Similarly, Nb127D01-mCherry and mito-GFP-127D01 co-localized with mitochondria (*Figure 1—figure supplement 2H*).

We next examined whether the position of the NanoTags affects recognition by the nanobodies. We generated H2B-mCherry, mito-mCherry, and CD8-mCherry with N-terminal, internal, or C-terminal NanoTags. When these vectors were co-transfected with NbVHH05-GFP or Nb127D01-GFP in S2*R*+ cells, GFP and mCherry co-localized with the expected cellular compartments (*Figure 1E–J*). Taken together, our data show that both NbVHH05 and Nb127D01 can be used as CBs to visualize and monitor NanoTagged proteins in their native surroundings.

### Detecting NanoTagged proteins by immunofluorescence

In addition to establishing CB-based detection of NanoTagged proteins in cells, we also explored detection by immunofluorescence. Nanobodies can be detected by either direct or indirect immunofluorescence. Direct immunofluorescence involves the use of fluorophore-conjugated nanobodies to detect the target protein(s). Indirect immunofluorescence involves recognition of the target by the nanobody, followed by detection of the nanobody recognized by a secondary fluorophore-conjugated antibody.

For direct immunofluorescence, we chemically conjugated NbVHH05 and Nb127D01 with fluorophores (NbVHH05-CF555 and Nb127D01-CF647). The success of these conjugations was confirmed by the detection of a fluorescent signal on an SDS-PAGE gel (*Figure 2—figure supplement 1*). Immunostaining with these fluorophore-conjugated nanobodies directly revealed VHH05- and 127D01-tagged proteins in S2*R*+ cells (*Figure 2A and A'*; *Figure 2—figure supplement 2A1, A2* ). Direct

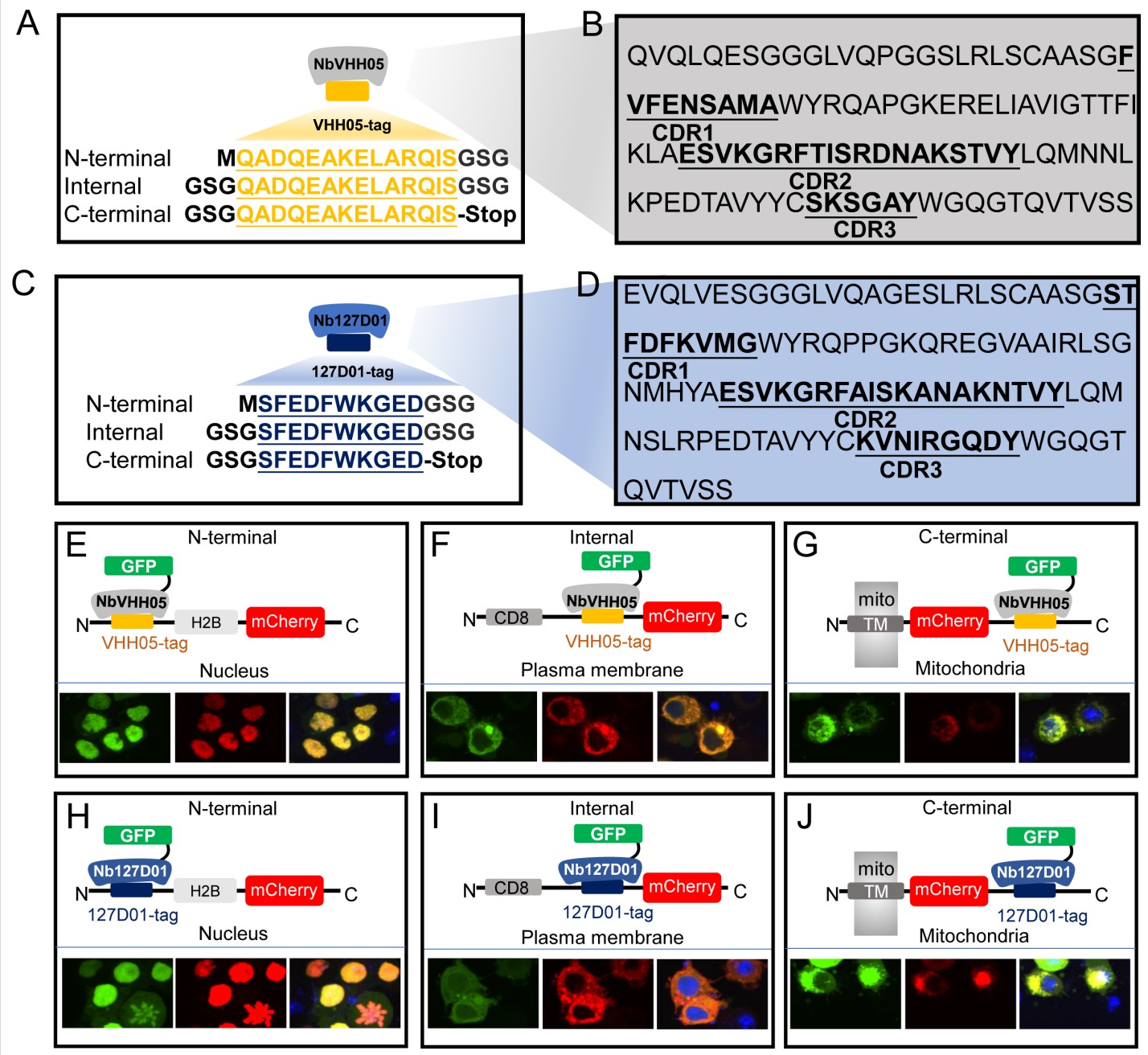

**Figure 1.** VHH05 and 127D01 NanoTag sequences and their corresponding nanobodies, and use of nanobodies as chromobodies. (**A and C**) VHH05 and 127D01 were inserted at the N-terminus, internally or at the C-terminus of a protein of interest (POI). GSG denotes the linker, M is the start codon, and Stop is the stop codon. (**B and D**) Nanobody sequences of NbVHH05 and Nb127D01. Bolded and underlined CDR1-3 corresponds to complementarity-determining regions (CDRs). (**E**) Co-transfection of pAW-actin5C-NbVHH05-GFP and pAW-actin5C-VHH05-H2B-mCherry into S2R+ cells. H2B is a nuclear protein. The right most panel is GFP, the center panel is mCherry, and the rightmost is the merged image. 4′,6-Diamidino-2-phenylindole (DAPI) staining shows the nuclei. (**F**) Co-transfection of pAW-actin5C-NbVHH05-GFP and pAW-actin5C-CD8-VHH05-mCherry into S2R+ cells. CD8 is a cell membrane protein. (**G**) Co-transfection of pAW-actin5C-NbVHH05-GFP and pAW-actin5C-mito-mCherry-VHH05 into S2R+ cells. Mito-mCherry-VHH05 contains a localization signal peptide for mitochondrial outer membrane targeting. (**H, I, and J**) Experiments are as in E, F, and G, except that pAW-actin5C-Nb127D01-GFP and pAW-actin5C-127D01-H2B-mCherry were co-transfected.

The online version of this article includes the following figure supplement(s) for figure 1:

**Figure supplement 1.** Identification of the 127D01 epitope.

**Figure supplement 2.** Schematic representation of the constructs and confocal images in S2R+ cells.

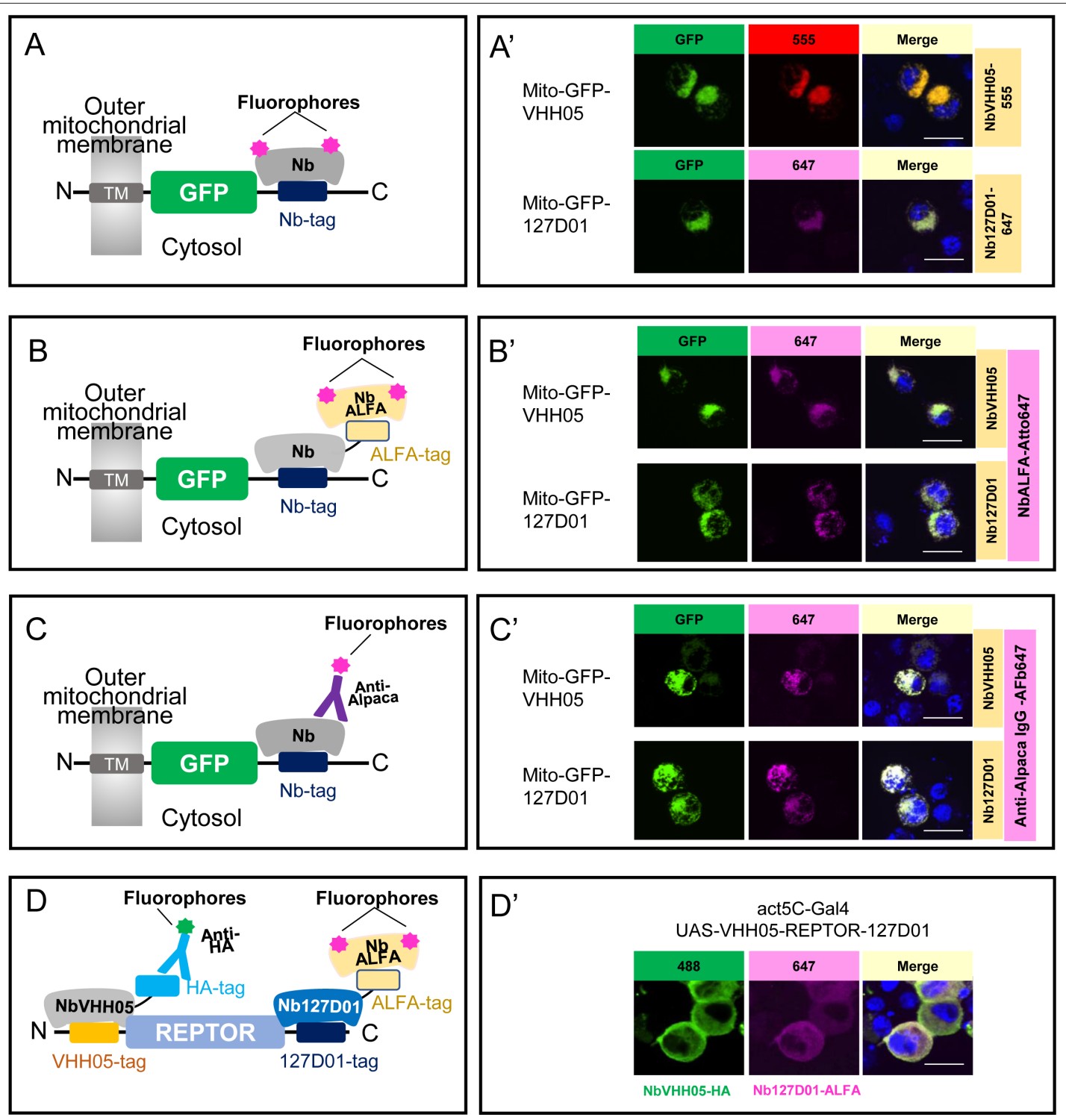

**Figure 2.** Using NbVHH05 and Nb127D01 for immunofluorescence. (**A**) Fluorophore-conjugated NbVHH05 or Nb127D01 recognizes VHH05- or 127D01-tagged fluorescence proteins. (**A'**) VHH05- or 127D01-tagged mito-GFP can be detected by the corresponding NbVHH05-555 or Nb127D01-647 in transfected S2R+ cells. 4',6-Diamidino-2-phenylindole (DAPI) staining shows the nuclei. (**B**) Schematic of nanobodies containing ALFA-tag as primary antibody and NbALFA as a secondary antibody. (**B'**) VHH05- or 127D01-tagged mito-GFP can be detected using the corresponding nanobodies in transfected S2R+ cells. (**C**) Schematic of fluorophore-conjugated anti-Alpaca IgG antibodies to detect VHH05- and 127D01-tagged proteins. NbVHH05 or Nb127D01 is used as primary antibodies and anti-Alpaca IgG as secondary antibody. (**C'**) VHH05- or 127D01-tagged mito-GFP can be detected using the corresponding nanobodies and anti-Alpaca IgG-647 in transfected S2R+ cells. (**D**) Schematic of using VHH05 and 127D01 for double tagging. N-,

*Figure 2 continued on next page*

*Figure 2 continued*

C-terminal of REPTOR contains VHH05 and 127D01. (**D'**) Co-staining NbVHH05 and Nb127D01 in S2*R*+ cells transfected with VHH05-REPTOR-127D01. Scale bars: 10 μm.

The online version of this article includes the following source data and figure supplement(s) for figure 2:

**Figure supplement 1.** Nanobody purification and fluorophore conjugation.

**Figure supplement 1—source data 1.** Raw data of fluorescence signals or coomassie brilliant blue staining for *Figure 2—figure supplement 1*.

**Figure supplement 2.** Different types of NbVHH05 and Nb127D01 and immunofluorescence examples.

**Figure supplement 3.** Test of potential interaction between VHH05 and 127D01.

**Figure supplement 3—source data 1.** Raw data of Western blot for *Figure 2—figure supplement 3b*.

immunostaining with fluorophore-conjugated nanobodies thus provides a simple and efficient method of detection. The reactivity of neither NbVHH05 nor Nb127D01 was affected by direct chemical conjugation. In addition, we used a fluorescent NbVHH05-AF555 prepared by site-specific Sortase labeling (*Figure 2—figure supplement 2A3*). For C-terminal Sortase-mediated labeling, the Sortase recognition motif (LPETG) is added to the C-terminus of the nanobody. Next, catalyzed by the Sortase, a fluorophore or biotin is added, using modified oligoglycine peptides such as GGG-fluorophore or GGG-biotin, to the nanobody-LPETG (*Cheloha et al., 2019*). These chemoenzymatic reactions proceed near-quantitatively and are highly site-specific. Unlike chemical modification, the sortase labeling procedure does not entail the risk of unwanted side reactions that might otherwise affect the physicochemical properties of the final product. Because each enzymatically modified nanobody carries a single substituent (fluorophore or biotin), quantitative comparisons are in principle possible, which would be more challenging when using direct chemical conjugation.

To test NbVHH05 and Nb127D01 in indirect immunofluorescence, we prepared bacterially purified NbVHH05 and Nb127D01, each fused with an ALFA-tag or HA-tag. We also prepared conditioned media that contained NbVHH05 or Nb127D01 tagged with the Fc portion of human IgG (hIgG) (*Figure 2—figure supplement 2B1*). Likewise, we could visualize NbVHH05-hIgG and Nb127D01-hIgG using fluorescently labeled anti-hIgG antibodies (*Figure 2—figure supplement 2B1*). Consistent with a previous report (*Götzke et al., 2019*), NbVHH05-ALFA and Nb127D01-ALFA can be visualized using anti-ALFA nanobodies conjugated to fluorophores (*Figure 2*; *Figure 2—figure supplement 2B2*). We could visualize NbVHH05-HA and Nb127D01-HA using fluorescently labeled anti-HA antibodies (*Figure 2—figure supplement 2B3*). Next, we determined whether ALFA-, HA-, or hIgG-tagged NbVHH05 and Nb127D01 could be used as the primary reagents for indirect immunostaining. We stained S2*R*+ cells transfected with mito-GFP-VHH05 or mito-GFP-127D01 vectors with the corresponding primary and secondary antibodies. In both cases, the GFP signal overlapped completely with the fluorescent signal from the secondary antibody (*Figure 2B'*; *Figure 2—figure supplement 2B*). NbVHH05 and Nb127D01 were obtained by immunization of an alpaca and llama, respectively (*Ling et al., 2019*; *Bradley et al., 2015*). We therefore examined whether they are both recognized by fluorophore-conjugated anti-alpaca IgG 647 which is also reactive with llama-derived nanobodies (or the VHH domain of llama IgG) (*Figure 2C*). The NanoTagged GFP and 647-fluorophore signals overlapped completely (*Figure 2C'*; *Figure 2—figure supplement 2B4*), indicating that commercially available secondary antibodies against llama are compatible with NbVHH05 and Nb127D01 immunostaining. In addition, we carried out indirect immunofluorescence using NbVHH05-biotin prepared by sortase labeling (*Cheloha et al., 2019*) and obtained similar results (*Figure 2—figure supplement 2B5*).

The availability of two different NanoTag-nanobody pairs opens the possibility for co-staining or co-detection with CBs. To test this, we generated proteins tagged with both NanoTags (VHH05- and 127D01-tags) and used NbVHH05 and Nb127D01 fused to HA-tag or ALFA-tag with corresponding secondary antibodies for detection. Importantly, these two tagging systems operate orthogonally, as no co-localization signal was observed in cells transfected with 127D01-GFP and with H2B-mCherry-VHH05 or mito-mCherry-VHH05 (VHH05-GFP with H2B-mCherry-127D01 or mito-mCherry-127D01) (*Figure 2—figure supplement 3A*). To further test co-detection, we inserted VHH05 at the N-terminus of the transcription factor REPTOR and 127D01 at its C-terminus (VHH05-REPTOR-127D01). When S2*R*+ cells were transfected with VHH05-REPTOR-127D01, the NbVHH05-HA signal (488-fluorophore) and the Nb127D01-ALFA signal (647-fluorophore) completely overlapped

(*Figure 2D and D'*). Our data show that immunostaining using both of the NanoTag-nanobody pairs can be multiplexed.

## Detection of NanoTagged proteins on immunoblots

Because the two nanobodies recognize small linear epitopes, we anticipated that these nanobodies might be useful for immunoblotting under denaturing conditions as already shown for VHH05 (*Ling et al., 2019*). We performed immunoblotting experiments with cell lysates containing VHH05- or 127D01-tagged H2B-mCherry. Using NbVHH05-ALFA and Nb127D01-ALFA as primary nanobodies, we detected a signal using NbALFA-HRP as the secondary antibody. Using purified nanobodies at high concentrations- produced some non-specific bands. This issue was resolved by reducing the nanobody concentration - (*Figure 3—figure supplement 1*). Next, we tested whether the two nanobodies could detect by immunoblotting proteins NanoTagged in different positions. NbVHH05 and Nb127D01 recognized proteins with internal, N- and C-terminal NanoTags on immunoblots (*Figure 3A*). To test whether increasing the number of NanoTags improved the sensitivity of detection, we generated vectors that express secreted GFP with 1x, 2x, or 3xVHH05 or 127D01 NanoTags at the C-terminus (*Figure 3B*). An N-terminal FLAG-tag was included in all constructs and used as the loading control. An increase in the number of NanoTags improved the sensitivity of detection using culture media that contain secreted GFP proteins (*Figure 3B*). Tagging of target proteins with more than one copy of a tag thus improves the sensitivity of detection.

Next, we tested whether the nanobodies against the VHH05 and 127D01 tags could be used for multiplexed immunoblots by double-tagging the Upd2 cytokine (VHH05-Upd2-127D01). As we needed to detect each nanobody in a specific manner, we tagged one nanobody with the ALFA tag and detected it using NbALFA and the other, with an hIgG. We expressed and prepared nanobody-hIgG from S2 cells and tested different concentrations of conditioned media on immunoblots (*Figure 3—figure supplement 2*). Very diluted conditioned media still produced a strong signal, even though the nanobody-hIgG was not purified or concentrated. After establishing working concentrations of hIgG-tagged nanobody conditioned media, we performed multiplexed immunostaining using Nb127D01-hIgG and NbVHH05-ALFA, or NbVHH05-hIgG and Nb127D01-ALFA, detected with anti-hIgG and NbALFA, respectively (*Figure 3C*). Upd2 undergoes fragmentation due to internal furin cleavage sites, which produced different bands on the immunoblot. A combination of hIgG-tagged nanobody and ALFA-tagged nanobody can thus be used for multiplexed immunoblotting (*Figure 3C*). In addition, we confirmed that a combination of NbVHH05-biotin and Nb127D01-hIgG also worked well for multiplexed immunoblotting (*Figure 3C*).

Another key application of antibodies is immunopurification of target proteins. To explore whether NbVHH05 and Nb127D01 can be used for immunopurification, we coated NbALFA resin with ALFA-tagged NbVHH05 or Nb127D01 nanobodies and used the modified resin to recover NanoTagged FLAG-GFP secreted in S2 cell culture media. FLAG-GFP-3xVHH05 and FLAG-GFP-3x127D01 were captured by NbVHH05-ALFA and Nb127D01-ALFA, respectively (*Figure 3D and E*). Protein A magnetic beads coated with Nb127D01-hIgG also successfully recovered FLAG-GFP-3x127D01 from S2 cell culture media (*Figure 3F*). We confirmed that the VHH05 and 127D01 systems do not cross-react (*Figure 2—figure supplement 3B*). NanoTag-based immunopurification with these nanobodies is thus effective.

## NanoTag trap as a method to alter protein localization

One of the many possible applications of nanobodies is to express them in cells or in vivo as fusion proteins localized to a particular subcellular location, in order to manipulate the localization of a Nano-Tagged POI. To test whether the NbVHH05/VHH05-tag and Nb127D01/127D01-tag can be used to alter localization, we constructed secreted GFP expression vectors that - had an N-terminal BiP signal peptide and a C-terminal VHH05- or 127D01-tag (BiP-GFP-VHH05 and BiP-GFP-127D01) (*Figure 4A and B*). We also constructed NbVHH05 and Nb127D01 with mCherry and KDEL endoplasmic reticulum (ER) retention signal (BiP-Nanobody-mCherry-KDEL), which should result in retention of the fusion protein in the ER (*Figure 4A and B*). When transfecting only BiP-GFP-VHH05 or BiP-GFP-127D01 into S2R+ cells, we did not observe GFP accumulation within cells, as the GFP proteins were actively secreted into the culture medium (*Figure 4C*). However, after co-transfecting BiP-NbVHH05-mCherry-KDEL with BiP-GFP-VHH05, or BiP-Nb127D01-mCherry-KDEL with BiP-GFP-127D01, we

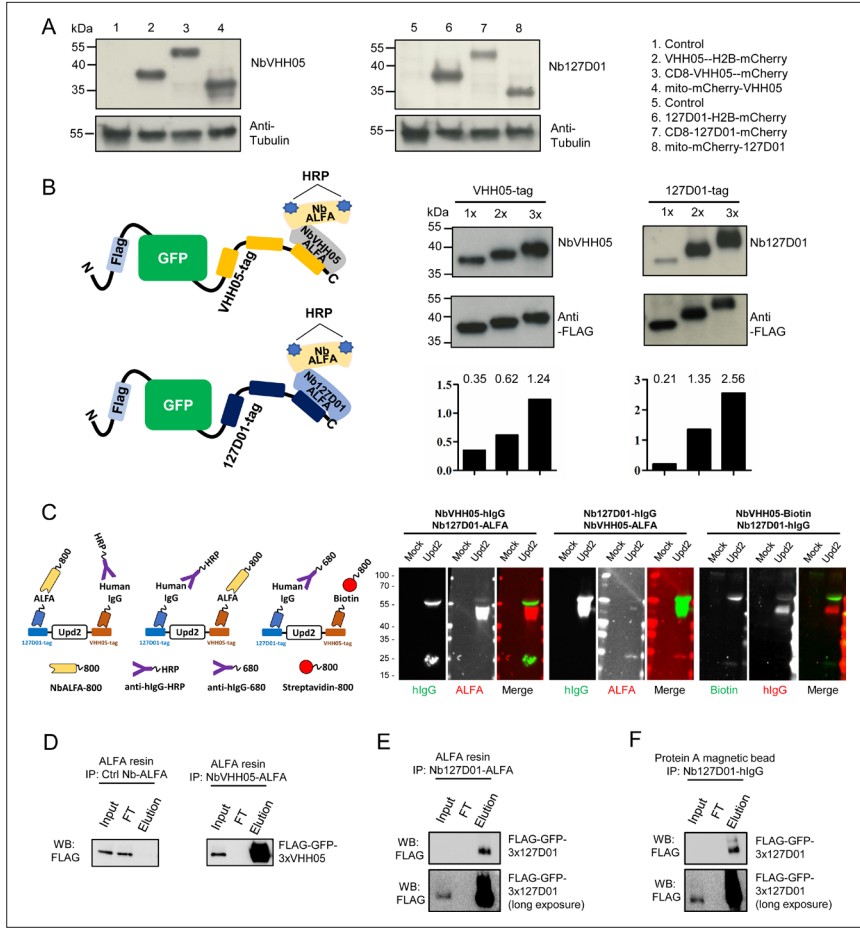

**Figure 3.** Detection of NanoTagged target proteins by western blotting and immunoprecipitation. (**A**) Lysates from S2R+ cells, transfected with different tagged plasmids (as used in *Figure 1*) or a mock control plasmid, were analyzed by SDS-PAGE and western blotting. The blot was developed with NbVHH05 and Nb127D01 followed by NbALFA-HRP or a mouse anti-tubulin primary antibody followed by anti-mouse IgG HRP. (**B**) Schematics depict VHH05- or 127D01-tagged secreted GFP proteins bound by NbVHH05-ALFA and Nb127D01-ALFA followed by NbALFA-HRP. Culture media from S2R+ cells transfected with secreted BiP-GFP-1xtag, BiP-GFP-2xtag, BiP-GFP-3xtag were used for the western blotting. Anti-FLAG antibody was used to show the GFP level. Histogram showing the relative gray value of anti-NbVHH05 or anti-Nb127D01 to anti-FLAG. (**C**) Western blots for S2R+ cell culture media containing double NanoTagged Upd2 protein: N- and C-terminus region of Upd2 contain VHH05 and 127D01, respectively, recognized by NbVHH05 or Nb127D01. The secondary antibodies were anti-hIgG-HRP, anti-ALFA-800, and Streptavidin-800. (**D**) Immunoprecipitation of FLAG-GFP-3xVHH05 using NbVHH05-ALFA and ALFA-resin. The control nanobody failed to capture FLAG-GFP-3xVHH05. (**E**) Immunoprecipitation of FLAG-GFP-3x127D01 using Nb127D01-ALFA and ALFA-resin. (**F**) Immunoprecipitation of FLAG-GFP-3x127D01 using Nb127D01-hIgG and Protein A magnetic bead.

The online version of this article includes the following source data and figure supplement(s) for figure 3:

**Source data 1.** Raw data of Western blot for *Figure 3*.

**Figure supplement 1.** Test of nanobody concentration gradient.

**Figure supplement 1—source data 1.** Raw data of Western blot for *Figure 3—figure supplement 1*.

**Figure supplement 2.** Rapid production of nanobodies in S2 cells for western blots.

**Figure supplement 2—source data 1.** Raw data of Western blot for *Figure 3—figure supplement 2*.

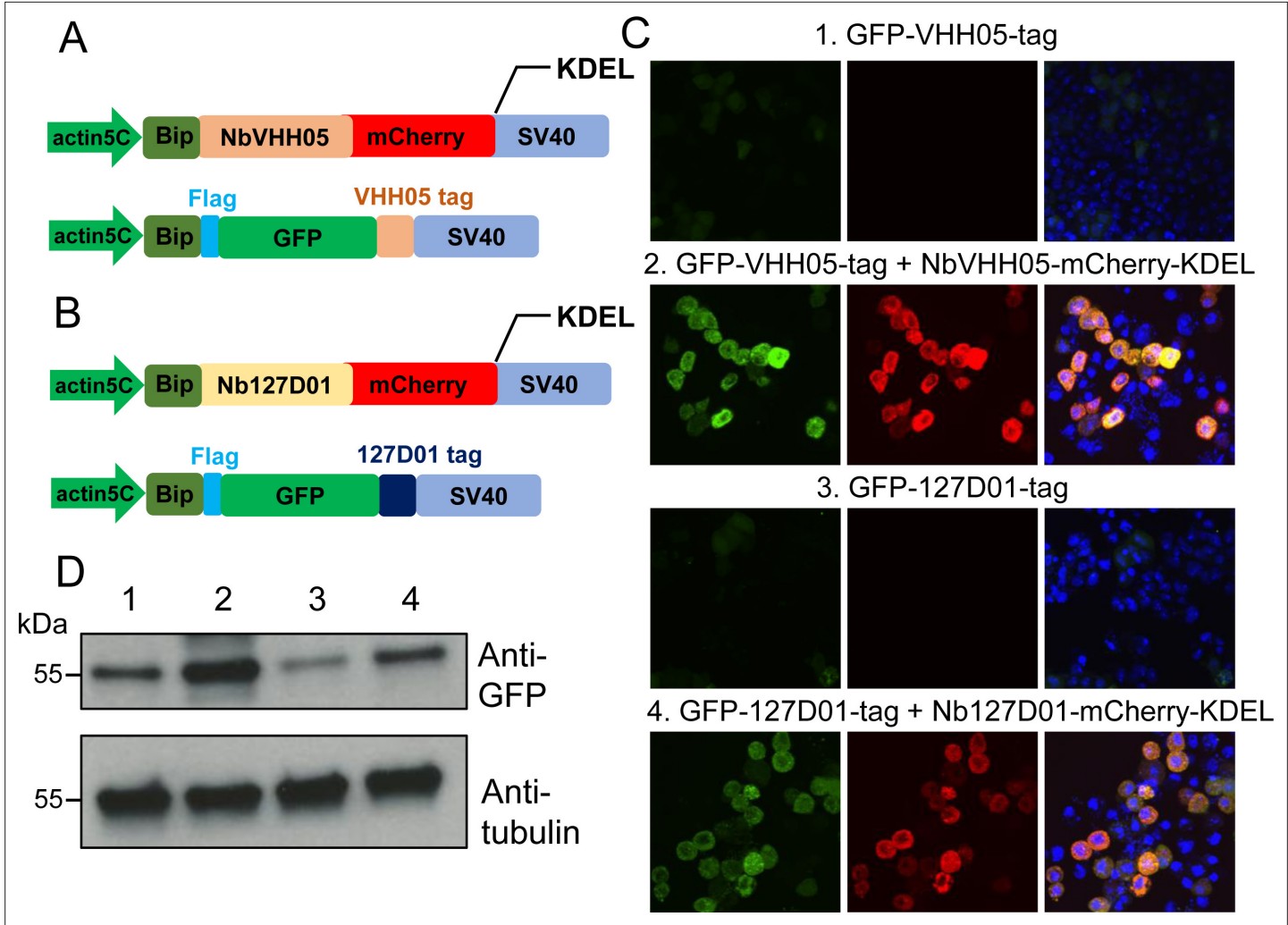

**Figure 4.** Nanobody-based system for altering localization of NanoTagged proteins. (**A and B**) Diagram showing the vectors used for the secreted protein trapping method. NbVHH05/Nb127D01 fused to mCherry contain KDEL and BiP signal peptide and is driven by the actin5C promoter. (**C**) Four independent cell transfection experiments were performed. In 1 and 3, only GFP-VHH05 or GFP-127D01 was transfected. In 2 and 4, NbVHH05-mCherry-KDEL with GFP-VHH05, or Nb127D01-mCherry-KDEL with GFP-127D01, were co-transfected. Images show the GFP and mCherry signal 48 hr after transfection. Nuclei are stained with 4',6-diamidino-2-phenylindole (DAPI). (**D**) Immunoblots of GFP and tubulin in cell lysates from transfections 1–4.

The online version of this article includes the following source data and figure supplement(s) for figure 4:

**Source data 1.** Raw data of Western blot for *Figure 4D*.

**Figure supplement 1.** Nanobody-based system for altering localization of NanoTagged proteins.

observed intracellular accumulation of GFP that co-localized with the mCherry signal. This shows that NanoTagged-GFP proteins directed to the secretory pathway can be trapped by an ER NanoTag trap (i.e. nanobodies with an ER retention signal) (*Figure 4C*). Indeed, cell lysates prepared from cells transfected with ER NanoTag trap showed more GFP signal on immunoblot, compared to controls lacking the ER NanoTag trap (*Figure 4D*).

We also prepared membrane-tethered nanobodies (membrane NanoTag trap) in order to re-localize cytoplasmic NanoTagged proteins to the membrane (*Figure 4—figure supplement 1A, B*). When S2*R*+ cells were co-transfected with CD8-NbVHH05-GFP/mito-mCherry-VHH05 or CD8-Nb127D01-GFP/mito-mCherry-127D01, mCherry co-localized with GFP on the cell membrane (*Figure 4—figure supplement 1C*). NanoTag traps targeted to particular cellular compartments can thus alter the subcellular distribution of NanoTagged proteins. The ability to do so may facilitate a variety of functional analyses.

## Assessing NanoTags in vivo

The successful use of the NbVHH05/VHH05-tag and Nb127D01/127D01-tag in cells prompted us to test them in vivo. We first constructed a series of UAS vectors with either a cytoplasmic or secreted version of the two nanobodies, tagged with GFP or HA (*Figure 5—figure supplement 1A*), and tested them in cells. As expected, when transfected together with the *pAct-Gal4* plasmid, the cytoplasmic expression vectors led to an expression of nanobodies detectable in S2*R*+ cell lysates, and the secretory expression vectors resulted in detection of nanobodies predominantly in the culture media (*Figure 5—figure supplement 1B, C*). Next, we generated transgenic flies carrying these UAS constructs. We did not expect NbVHH05 or Nb127D01 to interact with any fly endogenous proteins for lack of obvious sequence similarity between the *Drosophila* proteome and the amino acid sequence of the two tags (data not shown). To confirm this experimentally, we ubiquitously expressed NbVHH05 or Nb127D01 in vivo throughout development using *tubulin-Gal4*. The nanobodies were not toxic to flies; we readily obtained adult *tubulin-Gal4, UAS-Nanobody* flies and did not observe any detectable developmental defects or abnormalities (data not shown). To further test these constructs, we used fat body-specific *Lpp-Gal4* to drive NbVHH05 or Nb127D01 expression. Immunofluorescence showed that GFP- or HA-tagged NbVHH05 or Nb127D01 is expressed at readily detectable levels in fat body cells (*Figure 5A–D*). We also confirmed that nanobodies with the BiP signal peptide are secreted (*Figure 5E and F*).

To address whether NanoTagged POIs can be detected in vivo, we generated *UAS-VHH05-REPTOR-127D01* flies. A previous study has shown that in S2 cells, the transcription factor REPTOR is enriched in the cytoplasm under normal conditions but translocates into the nucleus upon rapamycin treatment (*Tiebe et al., 2015*). To test whether the changes in REPTOR localization can be detected using the nanobodies, we co-expressed NanoTagged REPTOR along with Nb127D01-GFP or NbVHH05-GFP specifically in adult enterocytes (ECs). In the absence of a NanoTagged POI, Nb127D01-GFP and NbVHH05-GFP were detected in both the cytoplasm and nuclei (*Figure 5G1*). In contrast, when co-expressed with NanoTagged REPTOR, the nanobody-GFP signals were enriched in the cytoplasm of ECs under normal food conditions (*Figure 5H and J*). Following rapamycin treatment, a stronger GFP signal was observed in nuclei. Similar changes were absent from ECs that express Nb127D01-GFP or NbVHH05-GFP alone (*Figure 5G1*). These results confirm that rapamycin treatment leads to translocation of REPTOR into the nucleus in vivo and provide further support for the idea that co-expression of NanoTagged POIs and nanobodies can visualize the subcellular location of a POI (*Figure 5H and J*).

Next, we checked whether NanoTagged proteins can be detected in vitro and in vivo using purified nanobodies. We first constructed several vectors that contain NanoTags at the N- or C-terminus of POIs and tested them in S2*R*+ cells. For secreted proteins, we replaced the endogenous signal peptide with the BiP signal peptide (*Figure 5—figure supplement 1D*). NanoTagged Akh, Dilp2, Dilp8, Pvf1, and Upd2 were readily detected in the culture media using NbVHH05 or Nb127D01 (*Figure 5—figure supplement 1E*). REPTOR and two isoforms of REPTOR-BP (REPTOR-BP-B and REPTOR-BP-C) could also be detected using NbVHH05 or Nb127D01 from transfected S2*R*+ cell lysates (*Figure 5—figure supplement 1F, G*). In addition to *VHH05-REPTOR-127D01* flies, we also generated two additional transgenic flies: *UAS-VHH05-REPTOR-BP-C-127D01* and *UAS-BiP-VHH05-Upd2-127D01* (a version of Upd2 with the BiP secretion signal and both NanoTags). We used *Myo1A*ᵗˢ to drive *UAS-VHH05-REPTOR-BP-C-127D01* and *UAS-BiP-VHH05-Upd2-127D01* and were able to detect the signal in adult midguts, using either NbVHH05 or Nb127D01 for detection by immunofluorescence (*Figure 5—figure supplement 2*). Taken together, these data indicate that both NbVHH05/VHH05-tag and Nb127D01/127D01-tag work well for in vivo imaging and immunostaining.

## CRISPR-mediated tagging of endogenous genes with NanoTags

In many cases, tagging endogenous proteins is preferable to UAS-based overexpression of tagged cDNAs, as UAS/Gal4-mediated expression can exceed physiological levels. Further, while many proteins have been tagged endogenously with GFP (*Morin et al., 2001*; *Sarov et al., 2016*; *Li-Kroeger et al., 2018*), tagging with smaller epitope tags may be preferable to minimize their structural impact. To tag endogenous genes with either VHH05-tag or 127D01-tag, we used a standard CRISPR-Cas9 targeted insertion method to tag fly proteins at their N- or C- terminus via the homology directed repair pathway (*Figure 6—figure supplement 1A, B, C*). To facilitate this approach, we first designed four universal vectors based on the scarless editing CRISPR knock-in

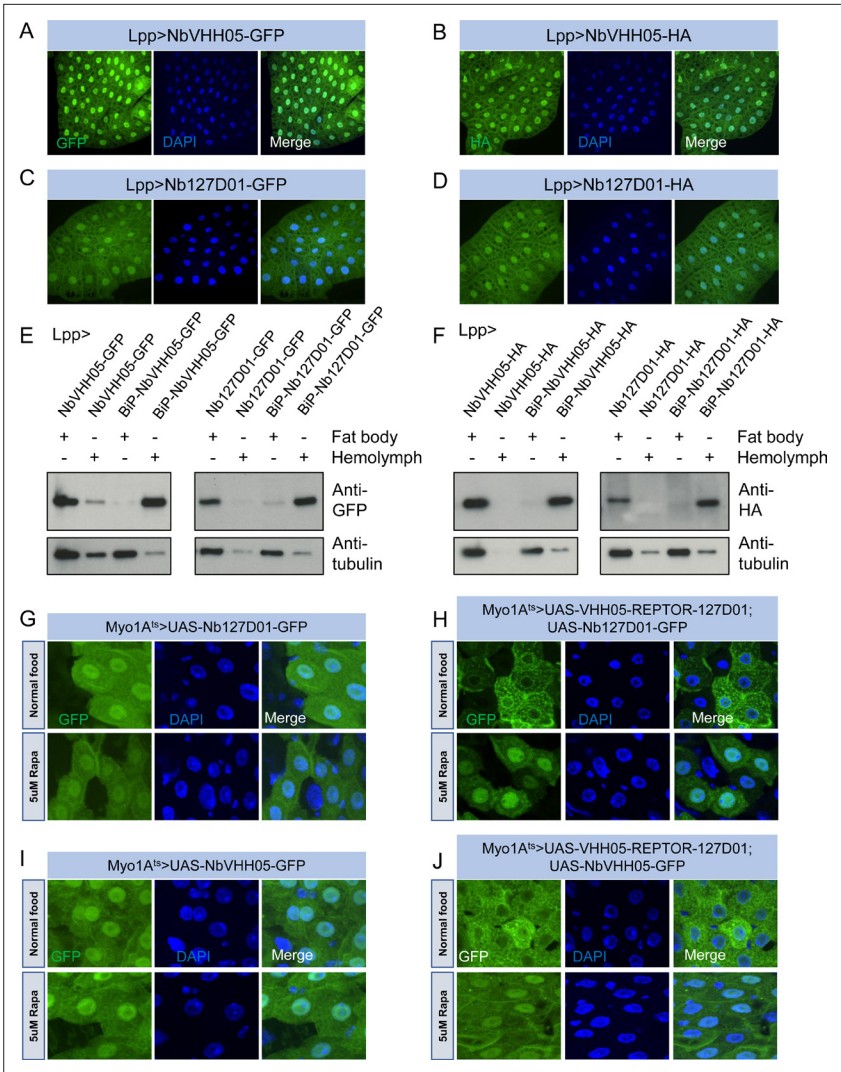

**Figure 5.** Nanobodies expression in vivo. (**A–D**) *Lpp-Gal4* drives fat body expression of *UAS-NbVHH05-GFP, UAS-NbVHH05-HA, UAS-Nb127D01-GFP,* or *UAS-Nb127D01-HA,* detected by GFP or anti-HA immunostaining. (**E and F**) Western blot detection of cytoplasmic and secreted GFP- or HA-tagged nanobodies. Lysates from fat body or hemolymph were tested by anti-GFP, anti-HA, and anti-tubulin antibodies. Cytoplasmic-expressed nanobodies: *UAS-NbVHH05-GFP, UAS-Nb127D01-GFP, UAS-NbVHH05-HA,* and *UAS-Nb127D01-HA.* Secreted-expressed nanobodies: *UAS-BiP-NbVHH05-GFP, UAS-BiP-Nb127D01-GFP, UAS-BiP-NbVHH05-HA,* and *UAS-BiP-Nb127D01-HA.* (**G–J**) Confocal images of *Drosophila* adult guts expressing 127D01-EGFP/VHH05-GFP and VHH05-REPTOR-127D01 with or without rapamycin (Rapa) treatment for 15 hr. REPTOR shuttles into the nucleus upon Rapa treatment. (**G and I**) as controls only express 127D01-EGFP or VHH05-GFP in the ECs. (**H**) combines 127D01-EGFP with VHH05-REPTOR-127D01 and (**J**) combines VHH05-EGFP with VHH05-REPTOR-127D01.

The online version of this article includes the following source data and figure supplement(s) for figure 5:

**Source data 1.** Raw data of Western blot for *Figure 5E and F*.

**Figure supplement 1.** Transgenic vector information and test in S2R+ cells.

**Figure supplement 1—source data 1.** Raw data of Western blot for *Figure 5—figure supplement 1*.

**Figure supplement 2.** Immunostaining of double NanoTag-labeled proteins.

(KI) approach (*Lamb et al., 2017*). Each vector contains five common features: the encoded Nano-Tags for N- or C-terminal tagging, *3xP3-dsRed-SV40* for identification of transformants, 5'/3' terminal repeats for piggyBac transposase recognition sequences, TTAA for piggyBac target sequence, and an EcoRI restriction site for cloning target locus homologous arms by Gibson assembly (*Figure 6—figure supplement 1A'*). We chose histone H2A variant (H2Av) as an example, because the expected

nuclear localization of H2Av should be easily visualized. We cloned the sequence 1 kb upstream and 1 kb downstream of the stop codon (TAA) into the donor vector (*Figure 6—figure supplement 2A*). A- sgRNA plasmid that targeted a seed sequence near the TAA of H2Av gene was injected into *yw; nos-cas9/CyO* embryos together with the donor plasmid (*Figure 6—figure supplement 1B'*). Positive transformants with red fluorescent eyes were outcrossed and successful KI events were confirmed by junction PCR and sequencing (*Figure 6—figure supplement 1C'*). Subsequently, 3xP3dsRed was excised using piggyBac transposase (*Figure 6—figure supplement 2B*). After sequence verification (*Figure 6—figure supplement 2C*), we immunostained midguts from H2Av-3x127D01 and H2Av-3xVHH05-expressing flies using Nb127D01-HA/NbVHH05-HA or NbVHH05-555/NbVHH05-biotin. As shown in *Figure 6—figure supplement 1D'* and *Figure 6—figure supplement 2D*, H2Av tagged with either NanoTag was clearly observed in the nucleus, demonstrating that this tagging method can be used effectively to engineer NanoTagged forms of POIs and study their localization and/or function at physiological expression levels.

In order to further test the sensitivity and the possibility of labeling secreted small peptides by the KI method, we selected *Dilp2* as an example. The above results showed that the double NanoTagged Dilp2 can be secreted into S2R+ culture medium and detected by western blot using NbVHH05 or Nb127D01 (*Figure 5—figure supplement 1F*). Here, we integrated 3x127D01 into the C-terminal of endogenous *Dilp2* using KI strategy (*Figure 6A, B, C*). After confirming the proper integration (*Figure 6D*), we compared Nb127D01 and Dilp2 antibody stainings in the larval brain, which revealed similar expression patterns (*Figure 6E*; *Park et al., 2014*). Since introducing tags to proteins can negatively affect their functions and a previous study has shown that *Dilp2* RNAi increases trehalose levels (*Broughton et al., 2008*), we examined the trehalose level of 127D01-tagged Dilp2 expressing flies. No difference was observed compared to control (*Figure 6G*), suggesting that that NanoTagged Dilp2 is functional. Furthermore, the change in Dilp2 signal in IPC cells and their neuronal projections in larval brain after starvation was easily detected by Nb127D01 immunostaining (*Figure 6G*). These results indicate that the CRISPR-mediated tagging of endogenous genes with NanoTags can be used for detecting small peptides with high sensitivity.

## Discussion

In this study, we characterized two NanoTags (VHH05 and 127D01) and their corresponding nanobodies (NbVHH05 and Nb127D01), for use in *Drosophila* for cellular and in vivo studies. We show that these two systems can be used for in vivo detection via chromobodies (CBs), re-localization, direct or indirect immunostaining, immunoblotting, and immunopurification. The observation that these nanobodies recognize the NanoTags on immunoblots is particularly useful, as only few nanobodies that recognize a defined amino acid sequence have been characterized as suitable for immunoblotting (*Cheloha et al., 2020*) The utility of this system is further enhanced by ease of purification of nanobodies from either bacterial cells or *Drosophila* cultured cells. Modification of these nanobodies with ALFA or HA tags facilitates detection. Installation of a human Fc portion on these nanobodies enables the use of anti-human IgG antibodies as secondary reagents for detection. In addition, we used chemical labeling or site-specific sortase labeling to prepare nanobodies labeled with fluorophores or biotin. We have thus developed reagents with broad applicability in *Drosophila* research and beyond.

The system described here has significant advantages over conventional antibodies or anti-GFP nanobodies. Nanobodies, unlike conventional antibodies, are easily encoded as a single open reading frame in the genomes of model organisms or cells. In addition, the small size of the NanoTags may be preferable in many cases to GFP, as GFP is bulky and may affect the function of the tagged protein. Also, the relatively long protein maturation time of GFP limit its use for imaging nascent proteins. Finally, the anti-GFP nanobody, as used in deGradFP (*Caussinus et al., 2011*), only poorly recognizes unfolded GFP.

We show how these short tags can be introduced into endogenous genes, using the scarless CRISPR KI editing approach. Given the small size of the tags, additional methods for genome modification deserve to be explored. For example, ssDNA-based CRISPR KI can be used to insert short sequences into a precise location in the genome (*Ling et al., 2017*). Another possibility is prime editing, which relies on pegRNAs to insert sequences smaller than 48 bp into a chosen genomic position (*Anzalone et al., 2019*; *Bosch et al., 2021*).

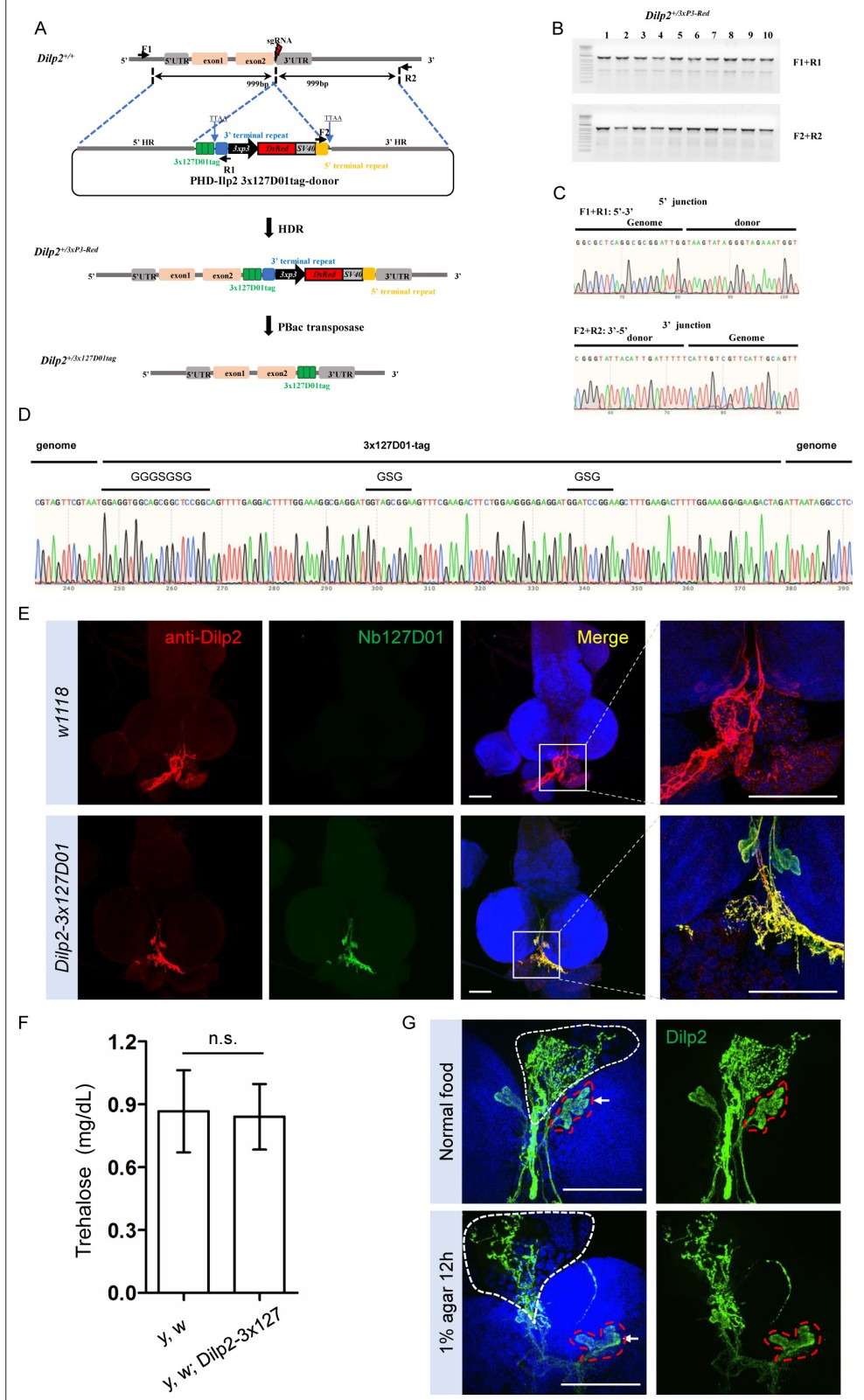

**Figure 6.** Integration of *3x127D01* into *Dilp2* shows a robust expression pattern in the brain. (**A**) Workflow and schematic representation of the *Dilp2* gene and the sgRNA targeting site. (**B**) PCR amplification was used to confirm the insertion. (**C**) Representative sequencing chromatogram of PCR products from the junction PCR. (**D**) Sequencing results of the DNA fragments showing 3x127D01-tag genome–donor integration. (**E**) Larval brains

*Figure 6 continued on next page*

*Figure 6 continued*

of *w1118* and *Dilp2−3x127D01* co-stained with anti-Dilp2 antibody and Nb127D01. Box demarcates Dilp2-expressing cells and Dilp2-positive neuronal projections. (**F**) Measurement of whole body trehalose concentration in *y,w* and *y,w; Dilp2−3x127D01* flies. Data are represented as mean ± SD and two-tailed t-tests were used to generate p values, n.s. indicates statistically non-significant. (**G**) Nanobody immunostaining showing different Dilp2 expression after starvation. Red outline demarcates Dilp2-expressing cells and white outline demarcates Dilp2-positive neuronal projections. Scale bars: 100 μm.

The online version of this article includes the following source data and figure supplement(s) for figure 6:

**Source data 1.** Raw data of agarose gel electrophoresis diagram for *Figure 6B*.

**Figure supplement 1.** Endogenous VHH05- or 127D01-tagging using CRISPR/Cas9.

**Figure supplement 1—source data 1.** Raw data of agarose gel electrophoresis diagram for *Figure 6—figure supplement 1C*.

**Figure supplement 2.** Schematic representation of the CRISPR/Cas9-mediated gene knock-in approach and the targeted integration of transgene constructs.

**Figure supplement 2—source data 1.** Raw data of agarose gel electrophoresis diagram for *Figure 6—figure supplement 2B*.

A key feature of the NanoTag/nanobody system is the ability to co-express the nanobody in vivo in the form of a CB or some other fusion. We show that this approach can be used to re-localize proteins. Expression in vivo also opens the doors to using versions of the deGrad system or other functional fusions to manipulate proteins in other ways (*Caussinus and Affolter, 2016*). Some S2*R*+ cells transfected with Nb127D01-GFP contained aggregates (data not shown). However, we did not observe aggregates in Nb127D01-mCherry transfected cells or transgenic Nb127D01-GFP flies, nor did we observe developmental defects or negative effects on viability following expression of the tags and/or nanobodies in vivo. The occasional observation of protein aggregation in cells more likely reflects the high level of expression when transfecting cells with exogenous expression constructs.

Given their versatility and in vivo applications, we anticipate that the two NanoTags and corresponding nanobodies will be useful to address many cell biological questions. The relative ease with which the system can be used for in vivo tagging as well as our demonstration that the system can be used in a number of ways for imaging (i.e. using CBs or direct or indirect immunofluorescence) are obvious assets. The NanoTag approach might be particularly useful for the new and growing need to spatially map cell clusters identified from scRNAseq studies (*Mohr et al., 2021*). Other potential applications for in vivo tagged proteins include chromosome immunoprecipitation, immunopurification of protein complexes, and inducing degradation of proteins that carry the tag (e.g. as shown for GFP in *Caussinus et al., 2011*; *Neumüller et al., 2012*). Another benefit of double labeling is that nanobody binding (for manipulation) may prohibit subsequent detection of the protein by the same nanobody; for example, when membrane localized Nanobody 1 is used to manipulate the localization of Protein X-NanoTag 1, Nanobody 1 cannot be used to visualize Protein X-NanoTag 1. This issue can be resolved with double NanoTagged proteins (NanoTag2-proteinX-NanoTag1), as Nanobody 2-Fluorophore should allow detection when membrane-localized Nanobody 1 is used as a trap.

Although we developed and validated the NanoTag system for use in *Drosophila*, the same approaches should be applicable to other systems. The mouse genome contains a predicted amino acid sequence with 100% identity to the epitope recognized by VHH05, which is not surprising, given that the tag sequence is identical to a segment of the human source protein, Ube6e. For 127D01, however, no proteins encoded by the mouse genome have peptides with more than 50% identity to the tag, suggesting that 127D01 could be used for murine studies. Moreover, there are no peptides with more than 50% identity to either tag in many species of research interest, including *E. coli*, *Saccharomyces cerevisiae*, *C. elegans*, *Arabidopsis thaliana*, *Oryza sativa* (rice), the mosquitoes *Aedes aegypti*, *Anopheles gambiae* and *Anopheles stephensi*, *Bombyx mori* (silkworm), *Tribolium castaneum* (red flour beetle), *Danaus plexippus* (monarch butterfly), and *D. rerio* (zebrafish), suggesting that both tags could be used in these organisms.

# Materials and methods

**Key resources table**

| Reagent type (species) or resource | Designation | Source or reference | Identifiers | Additional information |
|---|---|---|---|---|
| Antibody | Mouse monoclonal anti-a-Tubulin | Sigma-Aldrich | Cat# T5168; RRID: AB_477579 | WB (1:10,000) |
| Antibody | Mouse monoclonal anti-GFP | Invitrogen | Cat# A11120; RRID: AB_221568 | IF (1:300) |
| Antibody | Rabbit polyclonal anti-GFP | Molecular Probes | Cat# A-6455 RRID:AB_221570 | WB (1:10,000) |
| Antibody | Rat monoclonal anti-HA | Sigma-Aldrich | Cat# 3 F10; RRID: AB_2314622 | WB (1:10,000) IF (1:1000) |
| Antibody | Mouse monoclonal anti-FLAG M1 | Sigma-Aldrich | Cat# F3040; RRID: AB_439712 | WB (1:5000) |
| Antibody | Rabbit polyclonal anti-Dilp2 | *Park et al., 2014* | N/A | IF (0.5 µg/ml) |
| Other | NbALFA-HRP | NanoTag Biotechnologies | Cat# N1502-HRP | WB (1:5000) |
| Other | NbALFA-Atto647 | NanoTag Biotechnologies | Cat# N1502-At647N-L | IF (1:500) |
| Other | NbALFA-800CW | NanoTag Biotechnologies | Cat# N1502-Li800-L | WB (1:5000) |
| Antibody | Goat anti-alpaca IgG-647 | Jackson ImmunoResearch | Cat# 128-605-230 RRID: AB_2810930 | IF (1:500) |
| Antibody | Mouse monoclonal anti-HA-Alexa Fluor 488 | Thermo Fisher Scientific | Cat# A-21287 RRID: AB_2535829 | IF (1:1000) |
| Antibody | Goat anti-human IgG Fc-HRP | Thermo Fisher Scientific | Cat# A18829 RRID: AB_2535606 | WB (1:5000) |
| Antibody | Donkey anti-human IgG-DyLight 680 | Thermo Fisher Scientific | Cat# SA5-10130 RRID: AB_2556710 | WB (1:5000) |
| Peptide, recombinant protein | Streptavidin-DyLight 800 | Thermo Fisher Scientific | Cat# 21851 | WB (1:5000) |
| Peptide, recombinant protein | Streptavidin-Alexa Fluor 488 | Thermo Fisher Scientific | Cat# S32354 RRID: AB_2315383 | IF (1:500) |
| Other | NbVHH05-HA | This paper | N/A | WB (1:5000) IF (1:500) See *Note |
| Other | NbVHH05-ALFA | This paper | N/A | WB (1:5000) IF (1:500) See *Note |
| Other | NbVHH05-hIgG | This paper | N/A | WB (1:100) IF (1:20) See *Note |
| Other | NbVHH05-555 | This paper | N/A | IF (1:500) See *Note |
| Other | NbVHH05-biotin (sortagging) | This paper | N/A | IF (1:500) See *Note |
| Other | NbVHH05 –555 (sortagging) | This paper | N/A | IF (1:500) See *Note |
| Other | Nb127D01-HA | This paper | N/A | WB (1:5000) IF (1:500) See *Note |
| Other | Nb127D01-ALFA | This paper | N/A | WB (1:5000) IF (1:500) See *Note |
| Other | Nb127D01-hIgG | This paper | N/A | WB (1:100) IF (1:20) See *Note |
| Other | Nb127D01-647 | This paper | N/A | IF (1:500) See *Note |
| Peptide, recombinant protein | Phusion polymerase | New England Biolabs | Cat# M0530 | |
| Peptide, recombinant protein | Q5 polymerase | New England Biolabs | Cat# M0494 | |
| Peptide, recombinant protein | Taq polymerase | Clontech | Cat# TAKR001 | |

*Continued on next page*

*Continued*

| Reagent type (species) or resource | Designation | Source or reference | Identifiers | Additional information |
|---|---|---|---|---|
| Peptide, recombinant protein | EcoRI | New England Biolabs | Cat# R0101 | |
| Peptide, recombinant protein | XbaI | New England Biolabs | Cat# R0145 | |
| Peptide, recombinant protein | BglII | New England Biolabs | Cat# R0144 | |
| Peptide, recombinant protein | NheI | New England Biolabs | Cat# R3131 | |
| Peptide, recombinant protein | NsiI-HF | New England Biolabs | Cat# R3127 | |
| Peptide, recombinant protein | NcoI-HF | New England Biolabs | Cat# R3193 | |
| Peptide, recombinant protein | XhoI | New England Biolabs | Cat# R0146 | |
| Peptide, recombinant protein | BbsI | New England Biolabs | Cat# R0539 | |
| Peptide, recombinant protein | AarI | Thermo Fisher Scientific | Cat# ER1581 | |
| Peptide, recombinant protein | T4PNK | New England Biolabs | Cat# M0201 | |
| Peptide, recombinant protein | T4 DNA ligase | New England Biolabs | Cat# M0202 | |
| Peptide, recombinant protein | Fetal bovine serum | Sigma-Aldrich | Cat# A3912 | |
| Chemical compound, drug | Schneider's media | Thermo Fisher Scientific | Cat 21720–024 | |
| Chemical compound, drug | Penicillin-streptomycin | Thermo Fisher Scientific | Cat# 15070–063 | |
| Chemical compound, drug | ESF921 media | Expression Systems | Cat# 96–001 | |
| Peptide, recombinant protein | Proteinase K | Roche | Cat# 3115879001 | |
| Peptide, recombinant protein | RNase A | Thermo Fisher Scientific | Cat# EN0531 | |
| Peptide, recombinant protein | Protease and phosphatase inhibitor cocktail | Pierce | Cat# 78440 | |
| Chemical compound, drug | Trypsin inhibitor benzamidine | Sigma-Aldrich | Cat# 434760 | |
| Chemical compound, drug | Rapamycin | LC Laboratories | Cat# R-5000 | |
| Peptide, recombinant protein | HRP-Conjugated Streptavidin | Thermo Fisher Scientific | Cat# N100 | |
| Peptide, recombinant protein | Trehalase (prokaryote) | Megazyme | Cat# E-TREH | |
| Commercial assay or kit | Gibson assembly | New England Biolabs | Cat# E2611 | |
| Commercial assay or kit | NEBuilder HiFi assembly | New England Biolabs | Cat# E2621 | |
| Commercial assay or kit | Golden Gate Assembly | New England Biolabs | Cat# E1601 | |
| Commercial assay or kit | pJET-1.2 vector kit | Fermentas | Cat# K1231 | |
| Commercial assay or kit | QIAquick Gel Extraction Kit | Qiagen | Cat# 28706 | |
| Commercial assay or kit | QIAquick Spin Columns | Qiagen | Cat# 28115 | |
| Commercial assay or kit | Effectene | Qiagen | Cat# 301427 | |
| Commercial assay or kit | B-PER II Bacterial Protein Extraction Reagent | Thermo Fisher Scientific | Cat# 78260 | |
| Commercial assay or kit | Mix-n-Stain CF 555 Antibody Labeling Kit | Sigma-Aldrich | Cat# MX555S100 RRID: AB_10960067 | |
| Commercial assay or kit | Mix-n-Stain CF 647 Antibody Labeling Kit | Sigma-Aldrich | Cat# MX647S100 RRID: AB_10961766 | |
| Commercial assay or kit | Ni-NTA resin | EMD Millipore | Cat# 70691–3 | |
| Commercial assay or kit | PD-10 column | GE Healthcare | Cat# GE17-0851-01 | |
| Commercial assay or kit | Lysis buffer | Pierce | Cat# 87788 | |

*Continued on next page*

*Continued*

| Reagent type (species) or resource | Designation | Source or reference | Identifiers | Additional information |
|---|---|---|---|---|
| Commercial assay or kit | SDS sample buffer | Thermo Fisher Scientific | Cat# 39001 | |
| Commercial assay or kit | 4–20% polyacrylamide gel | Bio-Rad | Cat# 4561096 | |
| Commercial assay or kit | Enhanced chemiluminescence (ECL) reagents | Amersham | Cat# RPN2209 | |
| Commercial assay or kit | Enhanced chemiluminescence (ECL) reagents | Pierce | Cat# 34095 | |
| Commercial assay or kit | ALFA Selector ST resin | Nanotag Biotechnologies | Cat# N1511 | |
| Commercial assay or kit | Pierce IP lysis buffer | Thermo Fisher Scientific | Cat# 87787 | |
| Commercial assay or kit | Protein A magnetic beads | Bio-Rad | Cat# 1614013 | |
| Commercial assay or kit | Tetramethylbenzidine-containing solution | Thermo Fisher Scientific | Cat# N301 | |
| Commercial assay or kit | Glucose Hexokinase Reagents | Thermo Fisher Scientific | Cat# TR15421 | |
| Recombinant DNA reagent | pAW | Perrimon lab | N/A | See *Note |
| Recombinant DNA reagent | pWalium10 | DGRC | Cat# 1470 | |
| Recombinant DNA reagent | pMK-33GW | Perrimon lab | N/A | See *Note |
| Recombinant DNA reagent | pET-26b | Novagen | Cat# 69862 | |
| Recombinant DNA reagent | pQUASp-mCD8mCherry | Addgene | Cat# 46164 RRID: Addgene_46164 | |
| Recombinant DNA reagent | pBac (3xP3-gTc'v; pUb:lox-mYFP-lox-H2BmCherry) | Addgene | Cat# 119064 RRID: Addgene_119064 | |
| Recombinant DNA reagent | pcDNA4TO-mito-mCherry-10xGCN4_v4 | Addgene | Cat# 60914 RRID: Addgene_60914 | |
| Recombinant DNA reagent | PXL-IE1-EGFP-nos-Cas9 | *Xu et al., 2020* | N/A | |
| Recombinant DNA reagent | pScarlessHD-2xHA-DsRed | Addgene | Cat# 80822 RRID: Addgene_80822 | |
| Recombinant DNA reagent | pCFD3 | Addgene | Cat# 49410 RRID: Addgene_49410 | |
| Recombinant DNA reagent | pAW-NbVHH05-GFP | This paper Addgene | Cat# 171570 | |
| Recombinant DNA reagent | pAW-Nb127D01-GFP | This paper Addgene | Cat# 171571 | |
| Recombinant DNA reagent | pAW-NbVHH05-mCherry | This paper Addgene | Cat# 171572 | |
| Recombinant DNA reagent | pAW-Nb127D01-mCherry | This paper Addgene | Cat# 171573 | |
| Recombinant DNA reagent | pAW-H2B-mCherry-VHH05 | This paper | N/A | See *Note |
| Recombinant DNA reagent | pAW-mito-mCherry-VHH05 | This paper | N/A | See *Note |
| Recombinant DNA reagent | pAW-CD8-mCherry-VHH05 | This paper | N/A | See *Note |
| Recombinant DNA reagent | pAW-H2B-mCherry-127D01 | This paper | N/A | See *Note |
| Recombinant DNA reagent | pAW-mito-mCherry-127D01 | This paper | N/A | See *Note |
| Recombinant DNA reagent | pAW-CD8-mCherry-127D01 | This paper | N/A | See *Note |
| Recombinant DNA reagent | pAW-VHH05-H2B-mCherry | This paper | N/A | See *Note |
| Recombinant DNA reagent | pAW-CD8-VHH05-mCherry | This paper | N/A | See *Note |
| Recombinant DNA reagent | pAW-127D01-H2B-mCherry | This paper | N/A | See *Note |
| Recombinant DNA reagent | pAW-CD8-127D01-mCherry | This paper | N/A | See *Note |

*Continued on next page*

*Continued*

| Reagent type (species) or resource | Designation | Source or reference | Identifiers | Additional information |
|---|---|---|---|---|
| Recombinant DNA reagent | pAW-BiP-NbVHH05-mCherry-KDEL | This paper Addgene | Cat# 171574 | |
| Recombinant DNA reagent | pAW-BiP-Nb127D01-mCherry-KDEL | This paper Addgene | Cat# 171575 | |
| Recombinant DNA reagent | pAW-CD8-NbVHH05-GFP | This paper Addgene | Cat# 171576 | |
| Recombinant DNA reagent | pAW-CD8-Nb127D01-GFP | This paper Addgene | Cat# 171577 | |
| Recombinant DNA reagent | pAW-HGP-BiP-FLAG-GFP-VHH05 | This paper | N/A | See *Note |
| Recombinant DNA reagent | pAW-HGP-BiP-FLAG-GFP-2xVHH05 | This paper | N/A | See *Note |
| Recombinant DNA reagent | pAW-HGP-BiP-FLAG-GFP-3xVHH05 | This paper | N/A | See *Note |
| Recombinant DNA reagent | pAW-HGP-BiP-FLAG-GFP-127D01 | This paper | N/A | See *Note |
| Recombinant DNA reagent | pAW-HGP-BiP-FLAG-GFP-2x127D01 | This paper | N/A | See *Note |
| Recombinant DNA reagent | pAW-HGP-BiP-FLAG-GFP-3x127D01 | This paper | N/A | See *Note |
| Recombinant DNA reagent | pW10-UAS-BiP-127D01-Akh-VHH05 | This paper | N/A | See *Note |
| Recombinant DNA reagent | pW10-UAS-BiP-127D01-Dilp2-VHH05 | This paper | N/A | See *Note |
| Recombinant DNA reagent | pW10-UAS-BiP-127D01-Dilp8-VHH05 | This paper | N/A | See *Note |
| Recombinant DNA reagent | pW10-UAS-BiP-127D01-Pvf1-VHH05 | This paper | N/A | See *Note |
| Recombinant DNA reagent | pW10-UAS-127D01-REPTOR-bp-B-VHH05 | This paper | N/A | See *Note |
| Recombinant DNA reagent | pW10-UAS-127D01-REPTOR-bp-C-VHH05 | This paper | N/A | See *Note |
| Recombinant DNA reagent | pMT-HGP-v3-Nb127D01-hIgG | This paper Addgene | Cat# 171564 | |
| Recombinant DNA reagent | pMT-HGP-v3-NbVHH05-hIgG | This paper Addgene | Cat# 171565 | |
| Recombinant DNA reagent | pET-26b-Nb127D01-HA-His | This paper Addgene | Cat# 171566 | |
| Recombinant DNA reagent | pET-26b-NbVHH05-HA-His | This paper Addgene | Cat# 171567 | |
| Recombinant DNA reagent | pET-26b-Nb127D01-ALFA-His | This paper Addgene | Cat# 171568 | |
| Recombinant DNA reagent | pET-26b-NbVHH05-ALFA-His | This paper Addgene | Cat# 171569 | |
| Recombinant DNA reagent | pW10-UAS-NbVHH05-HA | This paper | N/A | See *Note |
| Recombinant DNA reagent | pW10-UAS-BiP-NbVHH05-HA | This paper | N/A | See *Note |
| Recombinant DNA reagent | pW10-UAS-Nb127D01-HA | This paper | N/A | See *Note |
| Recombinant DNA reagent | pW10-UAS-BiP-Nb127D01-HA | This paper | N/A | See *Note |
| Recombinant DNA reagent | pW10-UAS-NbVHH05-GFP | This paper | N/A | See *Note |
| Recombinant DNA reagent | pW10-UAS-BiP-NbVHH05-GFP | This paper | N/A | See *Note |
| Recombinant DNA reagent | pW10-UAS-Nb127D01-GFP | This paper | N/A | See *Note |
| Recombinant DNA reagent | pW10-UAS-BiP-Nb127D01-GFP | This paper | N/A | See *Note |
| Recombinant DNA reagent | pW10-UAS-127D01-REPTOR-VHH05 | This paper | N/A | See *Note |
| Recombinant DNA reagent | pW10-UAS-BiP-127D01-Upd2-VHH05 | This paper | N/A | See *Note |
| Recombinant DNA reagent | pW10-UAS-BiP-127D01-Akh-VHH05 | This paper | N/A | See *Note |
| Recombinant DNA reagent | pCFD3-H2Av-sgRNA | This paper | N/A | See *Note |
| Recombinant DNA reagent | pScarlessHD-C-3x127D01-H2Av-DsRed | This paper | N/A | See *Note |
| Recombinant DNA reagent | pScarlessHD-C-3xVHH05-H2Av-DsRed | This paper | N/A | See *Note |

*Continued on next page*

*Continued*

| Reagent type (species) or resource | Designation | Source or reference | Identifiers | Additional information |
|---|---|---|---|---|
| Recombinant DNA reagent | pScarlessHD-C-3x127D01-DsRed | This paper Addgene | Cat# 171578 | |
| Recombinant DNA reagent | pScarlessHD-C-3xVHH05-DsRed | This paper Addgene | Cat# 171580 | |
| Recombinant DNA reagent | pScarlessHD-N-3x127D01-DsRed | This paper Addgene | Cat# 171579 | |
| Recombinant DNA reagent | pScarlessHD-N-3xVHH05-DsRed | This paper Addgene | Cat# 171581 | |
| Cell line (*Drosophila melanogaster*) | S2R+ | DGRC | Cat# 150 RRID: CVCL_Z831 | FlyBase Report: FBtc0000150 |
| Cell line (*Drosophila melanogaster*) | ESF921-adapted S2 cells | Expression Systems | Cat# 94–005S | |
| Genetic reagent (*Drosophila melanogaster*) | w1118 | Perrimon lab | N/A | See *Note |
| Genetic reagent (*Drosophila melanogaster*) | y,v; P{nos- phiC31\int.NLS}X; P{CaryP} attP40 | Perrimon lab | N/A | See *Note |
| Genetic reagent (*Drosophila melanogaster*) | y,w; P{nos- phiC31\int.NLS}X; P{CaryP} attP2 | Perrimon lab | N/A | See *Note |
| Genetic reagent (*Drosophila melanogaster*) | y,w; nos-Cas9/CyO | Perrimon lab | N/A | See *Note |
| Genetic reagent (*Drosophila melanogaster*) | y,w; TM3, Sb/TM6,Tb | Perrimon lab | N/A | See *Note |
| Genetic reagent (*Drosophila melanogaster*) | yw; Gla/CyO | Perrimon lab | N/A | See *Note |
| Genetic reagent (*Drosophila melanogaster*) | yw; If/CyO; MKRS/TM6, Tb | Perrimon lab | N/A | See *Note |
| Genetic reagent (*Drosophila melanogaster*) | Myo1A-Gal4, tub-Gal80$^{ts}$ | Perrimon lab | N/A | See *Note |
| Genetic reagent (*Drosophila melanogaster*) | Lpp-Gal4 | Perrimon lab | N/A | See *Note |
| Genetic reagent (*Drosophila melanogaster*) | yw; UAS-NbVHH05-HA, w + attp2 | This paper | N/A | See *Note |
| Genetic reagent (*Drosophila melanogaster*) | yw; UAS-NbVHH05-HA, w + attp40 | This paper | N/A | See *Note |
| Genetic reagent (*Drosophila melanogaster*) | yw; UAS-Nb127D01-HA, w + attp2 | This paper | N/A | See *Note |
| Genetic reagent (*Drosophila melanogaster*) | yw; UAS-Nb127D01-HA, w + attp40 | This paper | N/A | See *Note |
| Genetic reagent (*Drosophila melanogaster*) | yw; UAS-NbVHH05-GFP, w + attp2 | This paper | N/A | See *Note |
| Genetic reagent (*Drosophila melanogaster*) | yw; UAS-NbVHH05-GFP, w + attp40 | This paper | N/A | See *Note |
| Genetic reagent (*Drosophila melanogaster*) | yw; UAS-Nb127D01-GFP, w + attp2 | This paper | N/A | See *Note |
| Genetic reagent (*Drosophila melanogaster*) | yw; UAS-Nb127D01-GFP, w + attp40 | This paper | N/A | See *Note |
| Genetic reagent (*Drosophila melanogaster*) | yw; UAS-BiP-NbVHH05-HA, w + attp2 | This paper | N/A | See *Note |
| Genetic reagent (*Drosophila melanogaster*) | yw; UAS-BiP-NbVHH05-HA, w + attp40 | This paper | N/A | See *Note |

*Continued on next page*

*Continued*

| Reagent type (species) or resource | Designation | Source or reference | Identifiers | Additional information |
|---|---|---|---|---|
| Genetic reagent (*Drosophila melanogaster*) | yw; UAS-BiP-Nb127D01-HA, w + attp2 | This paper | N/A | See *Note |
| Genetic reagent (*Drosophila melanogaster*) | yw; UAS-BiP-Nb127D01-HA, w + attp40 | This paper | N/A | See *Note |
| Genetic reagent (*Drosophila melanogaster*) | yw; UAS-BiP-NbVHH05-GFP, w + attp2 | This paper | N/A | See *Note |
| Genetic reagent (*Drosophila melanogaster*) | yw; UAS-BiP-NbVHH05-GFP, w + attp40 | This paper | N/A | See *Note |
| Genetic reagent (*Drosophila melanogaster*) | yw; UAS-BiP-Nb127D01-GFP, w + attp2 | This paper | N/A | See *Note |
| Genetic reagent (*Drosophila melanogaster*) | yw; UAS-BiP-Nb127D01-GFP, w + attp40 | This paper | N/A | See *Note |
| Genetic reagent (*Drosophila melanogaster*) | yw; UAS-BiP-VHH05-Upd2-127D01, w + attp40 | This paper | N/A | See *Note |
| Genetic reagent (*Drosophila melanogaster*) | yw; UAS-VHH05-REPTOR-127D01, w + attp40 | This paper | N/A | See *Note |
| Genetic reagent (*Drosophila melanogaster*) | yw; UAS-VHH05-REPTOR-BP-C127D01, w + attp40 | This paper | N/A | See *Note |
| Genetic reagent (*Drosophila melanogaster*) | w; H2Av-3xVHH05/TM3, Sb | This paper | N/A | See *Note |
| Genetic reagent (*Drosophila melanogaster*) | w; H2Av-3x127D01/TM3, Sb | This paper | N/A | See *Note |
| Strain, strain background (*Escherichia coli*) | TOP10 *Escherichia coli* | Invitrogen | Cat# C404010 | |
| Strain, strain background (*Escherichia coli*) | BL21 (DE3) *Escherichia coli* | New England Biolabs | Cat# C25271 | |
| Sequence-based reagent | All oligos | This paper | See **Supplementary file 1** | |
| Software, algorithm | Photoshop | Adobe | RRID:SCR_014199 | |
| Software, algorithm | ImageJ | NIH | RRID:SCR_003070 | |
| Software, algorithm | Excel | Microsoft | RRID:SCR_016137 | |
| Software, algorithm | GraphPad Prism6 | GraphPad | RRID:SCR_002798 | |
| Other | Joystick Micromanipulator | NARISHIGE | Cat# MN-151 | |
| Other | FemtoJet Microinjector | Eppendorf | Cat# LV41365120 | |
| Other | Garfunkel Nikon Ti2 Spinning Disk | Nikon | N/A | |
| Other | Kimble Kontes pellet pestles | Millipore | Cat# Z359947 | |
| Other | Immobilon-P polyvinylidene fluoride (PVDF) membrane | Millipore | Cat# IPVH00010 | |
| Other | ChemiDoc MP imaging system | Bio-Rad | Cat# 17001402 | |
| Other | Kodak M35 X-OMAT Automatic Processors | KODAK | Cat# RT-KP-M35A | |
| Other | Hyperfilm ECL | Amersham | Cat# GE28-9068-35 | |
| Other | 4',6-Diamidino-2-phenylindole (DAPI) | Thermo Fisher Scientific | Cat# D1306 RRID: AB_2629482 | (1 µg/ml) |

*Note: Further information and requests for resources and reagents used in this paper should be directed to and will be fulfilled by the Lead Contact, Norbert Perrimon (perrimon@genetics.med.harvard.edu.). Transgenic flies used to express these two nanobodies and plasmids used to express and prepare nanobodies, which have been submitted to public reagent resource centers, Bloomington Drosophila Stock Center, Drosophila Genomics Resource Center and Addgene.

## Plasmid construction

Four types of vectors were used in this study: (1) pAW, that contains the fly actin5C promoter; (2) pWalium10, a UAS/Gal4 vector that contains a UAS promoter and mini-white selection marker (DGRC, 1470); (3) pMT (pMK-33GW, Ram Viswanatha); and (4) pET-26b (Novagen 69862). pAW-HGP-sortase is a pAW vector derivative with N-terminal BiP signal peptide/FLAG-tag, AarI-cloning site for Gibson or HiFi assembly, C-terminal purification tag with Sortase-tag and His-tag, and heat shock promoter-GFP-T2A-PuroR for stable cell line generation. pMT-HGP-v3 is a pMT vector derivative with N-terminal BiP signal peptide/FLAG-tag, AarI-cloning site for Gibson or HiFi assembly, C-terminal purification tag with Avi-tag and His-tag, and heat shock promoter-GFP-T2A-PuroR for stable cell line generation. pET-26b-Nb-GGA is a pET-26b derivative with BsaI Golden Gate Assembly cloning site flanked by pelB signal sequence and C-terminal ALFA- and His-tags.

Plasmid DNAs were constructed and amplified using standard protocols. Briefly, plasmids were linearized by restriction enzymes as described by the commercial vendor. PCR fragments were amplified using Phusion polymerase (New England Biolabs [NEB], M0530) or Q5 polymerase (NEB, M0494). Linearized plasmids and PCR fragments were gel purified using QIAquick columns (Qiagen, #28115) and joined using either Gibson assembly (NEB, E2611) or NEBuilder HiFi assembly (NEB, E2621). Reactions were transformed into chemically competent TOP10 *E. coli* (Invitrogen, C404010), plated and selected on lysogeny broth-agar plates with ampicillin (100 µg/ml) or kanamycin (50 µg/ml). Colony PCR was performed using Takara Taq polymerase (Clontech, TAKR001C). Plasmid DNA was isolated from cultured bacteria using QIAprep Spin Miniprep Kit (Qiagen, 27104). Plasmid sequences were confirmed by Sanger sequencing performed at the Dana-Farber/Harvard Cancer Center DNA Resource Core or Genewiz. Primers sequences are listed in *Supplementary file 1*.

## pW10-nanobody variants

All UAS constructs were cloned into the pWalium10 (pW10) vector. For pW10-NbVHH05-HA, pW10-Nb127D01-HA, pW10-NbVHH05-GFP, pW10-Nb127D01-GFP, pW10-BiP-NbVHH05-HA, pW10-BiP-Nb127D01-HA, pW10-BiP-NbVHH05-GFP, and pW10-BiP-Nb127D01-GFP, pW10 was first linearized with EcoRI (NEB, R0101) and XbaI (NEB, R0145). NbVHH05 or Nb127D01 was cloned from pHEN6-VHH05 or pHEN6-127D01 (Ploegh lab). GFP was amplified from PXL-IE1-EGFP-nos-Cas9 (*Xu et al., 2020*). PCR fragments were joined together with the digested pW10 backbone by Gibson assembly. HA and BiP were incorporated by adding overhanging sequences to the primers. For pW10-BiP-VHH05-Akh-127D01, pW10-BiP-VHH05-Dilp2-127D01, pW10-BiP-VHH05-Dilp8-127D01, pW10-BiP-VHH05-Pvf1-127D01, and pW10-BiP-VHH05-Upd2-127D01, pW10 was first digested with EcoRI and XbaI. A BiP-VHH05-127D01 fragment was made by annealing two oligos and ligated into the linearized pW10 to generate the intermediate vector, pW10-BiP-VHH05-BglII-127D01. Next, PCR fragments of *Akh, Dilp2, Dilp8, Pvf1,* and *Upd2* were amplified from fly cDNAs and inserted into pW10-UAS-BiP-VHH05-127D01 linearized by BglII (NEB, R0144) by Gibson assembly. For pW10-127D01-REPTOR-bp-B-VHH05, pW10-127D01-REPTOR-bp-C-VHH05, and pW10-127D01-REPTOR-VHH05, pW10-BiP-VHH05-127D01 was digested with EcoRI and BglII to remove BiP-VHH05. VHH05-tag was introduced N-terminal of PCR fragments of *REPTOR-bp-B, REPTOR-bp-C* and *REPTOR* through overhanging primers. PCR fragments were inserted into pW10-127D01 (EcoRI and BglII digested) by Gibson assembly.

## pAW-nanobody/NanoTagged fluorophore variants

For pAW-NbVHH05-GFP, pAW-Nb127D01-GFP, pAW-NbVHH05-mCherry, pAW-Nb127D01-mCherry, pAW-H2B-mCherry-VHH05, pAW-H2B-mCherry-127D01, pAW-mito-mCherry-VHH05, pAW-mito-mCherry-127D01, pAW-CD8-mCherry-VHH05, pAW-CD8-mCherry-127D01, pAW-VHH05-H2B-mCherry, pAW-127D01-H2B-mCherry, pAW-BiP-NbVHH05-mCherry-KDEL, pAW-BiP-Nb127D01-mCherry-KDEL, pAW-CD8-NbVHH05-GFP, and pAW-CD8-Nb127D01-GFP, pAW was first linearized with NheI (NEB, R3131) and XbaI. NbVHH05-GFP and Nb127D01-GFP were amplified from pW10-NbVHH05-GFP and pW10-Nb127D01-GFP. *CD8, mCherry*, and *CD8-mCherry* were cloned from pQUASp-mCD8mCherry (Addgene, #46164). *H2B-mCherry* was cloned from pBac (3xP3-gTc'v; pUb:lox-mYFP-lox-H2BmCherry) (Addgene, #119064). *mito-mCherry* was cloned from pcDNA4TO-mito-mCherry-10xGCN4_v4 (Addgene, #60914). BiP-NbVHH05 or BiP-Nb127D01 fragments were cloned from pW10-UAS-BiP-NbVHH05-GFP, pW10-UAS-BiP-Nb127D01-GFP. PCR fragments were

joined together with the digested pAW backbone by Gibson assembly. VHH05-, 127D01-tags, or KDEL sequences were incorporated in the N- or C-terminus by adding overhanging sequences to the primers. For pAW-CD8-VHH05-mCherry and pAW-CD8-127D01-mCherry, we first digested pAW-CD8-mCherry-VHH05 with NheI to obtain a linearized pAW-CD8 backbone. mCherry was amplified from pQUASp-mCD8mCherry. VHH05-tag or 127D01-tag was introduced into the N-terminal of PCR fragments with overhanging primer sequences. The NanoTag-mCherry PCR fragment was inserted into pAW-CD8 by Gibson assembly.

## pAW-HGP-BiP-GFP with 1x/2x/3x NanoTags

For pAW-HGP-BiP-GFP-VHH05, pAW-HGP-BiP-GFP-2xVHH05, pAW-HGP-BiP-GFP-3xVHH05, pAW-HGP-BiP-GFP-127D01, pAW-BiP-HGP-GFP-2x127D01, pAW-HGP-BiP-GFP-3x127D01, pAW-HGP-sortase was first linearized by AarI (Thermo Scientific ER1581). N-terminal GFP was amplified by PCR. C-terminal GFP with 1x/2x/3x NanoTags were prepared by gene synthesis (Twist Bioscience). PCR fragment and C-terminal GFP with NanoTags were joined together with the linearized pAW-HGP-sortase by NEBuilder HiFi assembly.

## pMT-HGP-V3-Nb127D01-hIgG, pMT-HGP-V3-NbVHH05-hIgG

For cloning of the inducible Nb127D01-hIgG and NbVHH05-hIgG expression vectors, the pMT-HGP-v3 vector was linearized by both AarI and NsiI-HF (NEB, R3127). hIgG Fc region was amplified from pCER243 (a gift of Aaron Ring at Yale). NbVHH05/Nb127D01 and hIgG PCR fragments were inserted into pMT-HGP-v3 (digested by AarI and NsiI-HF) by NEBuilder HiFi assembly.

## pET-26b-Nb127D01-ALFA-His, pET-26b-NbVHH05-ALFA-His, pET-26b-Nb127D01-HA-His, pET-26b-NbVHH05-HA-His

For cloning pET-26b-Nb127D01-ALFA-His and pET-26b-NbVHH05-ALFA-His, NbVHH05/Nb127D01 PCR fragments were cloned into pET-26b-Nb-GGA by BsaI-Golden Gate Assembly (NEB, E1601). To make pET-26b-Nb127D01-HA-His and pET-26b-NbVHH05-HA-His, NbVHH05/Nb127D01 PCR fragments with HA tag were cloned into pET-26b (NcoI, NEB, R3193 and XhoI, NEB, R0146 digested) by HiFi assembly.

## pScarless NanoTag vectors: pScarlessHD-C-3x127D01-DsRed, pScarlessHD-C-3xVHH05-DsRed, pScarlessHD-N-3x127D01-DsRed, pScarlessHD-N-3xVHH05-DsRed

3x127D01 or 3xVHH05 were cloned from pAW-HGP-BiP-GFP-3x127D01 or pAW-HGP-BiP-GFP-3xVHH05. 3xP3-DsRed was cloned from pScarlessHD-2xHA-DsRed (Addgene, 80822). Then, 3x127D01 or 3xVHH05 and 3xP3-DsRed PCR fragments were inserted into pScarlessHD-2xHA-DsRed backbone (EcoRI digested) by Gibson assembly.

## pScarlessHD-C-3x127D01-H2Av-DsRed, pScarlessHD-C-3xVHH05-H2Av-DsRed

pScarlessHD-C-3x127D01-DsRed and pScarlessHD-C-3xVHH05-DsRed were digested with EcoRI to obtain the pScarlessHD backbone and 3xP3-DsRed fragment. Upstream and downstream sequences of TAA at the C-terminus of the *H2Av* gene were cloned from *yw; nos-Cas9/Cyo* flies. Then, 3xP3-DsRed, upstream and downstream fragments were inserted into pScarlessHD backbone by Gibson assembly.

## pCFD3-H2Av-sgRNA

pCFD3 (Addgene, #49410) was digested with BbsI (NEB, R0539). sgRNA oligos following phosphorylation were annealed by T4PNK (NEB, M0201) and inserted into a digested pCFD3 backbone by T4 DNA ligase (NEB, M0202).

## Cell transfection

*Drosophila* S2*R*+ cells (DGRC, 150) were cultured at 25°C in Schneider's media (Thermo Fisher Scientific, 21720–024) with 10% fetal bovine serum (Sigma, A3912) and 50 U/ml penicillin-streptomycin

(Thermo Fisher Scientific, 15070–063). S2R+ cells were transfected using Effectene (Qiagen, 301427) following the manufacturer's instructions. A total of 200 ng of plasmid DNA per well was transfected in 24-well plates. The culture medium was replaced 24 hr after transfection.

To produce secreted GFP proteins with 1x/2x/3x NanoTags and nanobodies with hIgG, ESF921-adapted S2 cells (Expression Systems, 94–005S) were transfected using PEI transfection methods and cultured in protein-free ESF921 media (Expression Systems, 96–001). For six-well plate transfection, 2.4 µg of DNA and 7.4 µg of PEI were mixed in 300 µl of ESF921 media for 15–30 min and added to 2.7 ml of S2 cell culture (2E6 cells/ml, final cell density). Alternatively, 50 ml suspension cells cultured in 250 ml flask were transfected using the same proportion. On day 2 from transfection, protein expression was induced with 700 µM CuSO4 for 5 days. Cleared conditioned media were collected after centrifugation and directly used for immunostaining, western blot, and immunoprecipitation.

## Fly strains and generating transgenic flies

Fly husbandry and crosses were performed under standard conditions at 25°C. Injections were carried in-house. Fly strains used to generate transgenic lines were attP lines: attP40 (*y,v; P{nos- phiC31\int.NLS}X; P{CaryP}attP40*) and attP2 (*y,w; P{nos- phiC31\int.NLS}X; P{CaryP}attP2*). Fly strains used to generate KI lines were *y,w; nos-Cas9/CyO*. For balancing chromosomes, fly stocks *y,w; TM3, Sb/ TM6,Tb, yw; Gla/CyO, yw; If/CyO; MKRS/TM6,Tb* were used. All lines recovered were homozygous viable. Fly stocks are from the Perrimon fly stock unless stated otherwise. Other fly stocks used in this study were *Lpp-Gal4* (Hong-Wen Tang) and *Myo1A-Gal4, tub-Gal80$^{ts}$* (Afroditi Petsakou).

Transgenic flies generated in this study are as follows: yw; UAS-NbVHH05-HA, w + attP2 yw; UAS-NbVHH05-HA, w + attP40 yw; UAS-Nb127D01-HA, w + attP2
yw; UAS-Nb127D01-HA, w+ attP40 yw; UAS-NbVHH05-GFP, w+ attP2
yw; UAS-NbVHH05-GFP, w+ attP40 yw; UAS-Nb127D01-GFP, w+ attP2
yw; UAS-Nb127D01-GFP, w+ attP40 yw; UAS-BiP-NbVHH05-HA, w+ attP2
yw; UAS-BiP-NbVHH05-HA, w+ attP40 yw; UAS-BiP-Nb127D01-HA, w+ attP2
yw; UAS-BiP-Nb127D01-HA, w+ attP40
yw; UAS-BiP-NbVHH05-GFP, w+ attP2
yw; UAS-BiP-NbVHH05-GFP, w+ attP40 yw; UAS-BiP-Nb127D01-GFP, w+ attP2
yw; UAS-BiP-Nb127D01-GFP, w+ attP40 yw; UAS-BiP-VHH05-Upd2-127D01, w+ attP40
yw; UAS-VHH05-REPTOR-127D01, w+ attP40
yw; UAS-VHH05-REPTOR-BP-C-127D01, w+ attP40
w; H2Av-3xVHH05/TM3, Sb w; H2Av-3x127D01/TM3, Sb

Transgenic flies were generated by phiC31 integration of attB-containing plasmids into either attP40 or attP2 landing sites. Briefly, plasmid DNA was purified on QIAquick columns and eluted in injection buffer (100 µM NaPO₄, 5 mM KCl) at a concentration of 200–400 ng/µl. Plasmid DNA was injected into ~100 fertilized embryos (*y,v nos-phiC31int; attP40* or *y,w nos-phiC31int; attP2*) through microinjection handle (NARISHIGE, Japan) with pressure control (FemtoJet, Eppendorf). The progeny was outcrossed to screen for transgenic founder progeny and the UAS insertions were isolated by screening for white+ eye color.

To generate KI flies, donor and sgRNA plasmids were mixed together and were injected into ~100 fertilized embryos (*y,w; nos-Cas9/CyO*). Transformed lines were isolated using a DsRed marker. To remove the DsRed cassette, transformed lines were crossed to a line expressing PBac transposase (BL #8285). Resulting lines were sequenced to confirm the insertion of 3x127D01-tag or 3xVHH05-tag to the C-terminal of the *H2Av* or *Dilp2* gene.

## Targeted integration analysis

Genomic DNA was extracted from the DsRed2-positive G1 adult animals by standard sodium dodecyl sulfate (SDS) lysis-phenol buffer after incubation with proteinase K (Roche, 3115879001), followed by RNase A (Thermo Fisher Scientific, EN0531) treatment and purification. The 5'- and 3'-end junction fragments at the integration event were cloned separately and sequenced. PCR conditions included 2 min of denaturation at 95°C; 30 cycles of 1 min at 95°C, 30 s at 55°C and 1 min at 72°C; followed by a final extension at 72°C for 10 min by using Takara Taq polymerase (Clontech, #TAKR001). To confirm the removal of the DsRed cassette, the genomic DNA was extracted by the same method and PCR conditions included 2 min of denaturation at 95°C; 30 cycles of 1 min at 95°C, 30 s at 55°C and 30 s at

72°C; followed by a final extension at 72°C for 10 min. PCR products were subcloned into the pJET-1.2 vector (Fermentas, #K1231) and sequenced. Primer sequences are listed in the *Supplementary file 1*.

## Rapamycin treatment

For rapamycin treatment, rapamycin (LC Laboratories, R-5000) was dissolved in DMSO and mixed into the media when preparing food vials. The rapamycin dose was 50 μM.

## Nanobody purification

Nanobody expression vectors with C-terminal ALFA-tag or HA-tag for bacterial expression and purification were cloned into pET-26b-Nb-GGA or pET-26b by Golden-Gate assembly or Gibson cloning, respectively. After transformation with BL21 (DE3) competent cells (NEB, C25271), overnight cultures were inoculated in LK media and incubated at 37°C until the OD600 approached ~0.6. Then, 0.1 mM IPTG was added and cultures were placed at 15°C for overnight. After centrifugation, the cell pellet was collected and resuspended in B-PER II Bacterial Protein Extraction Reagent (Thermo Scientific, 78260). After 30 min incubation with rotation, an additional 19× volume of TBS was added and further incubated with rotation. Cleared supernatants were collected after centrifugation at 20,000*g* and 4 °C for 15 min and used for His-tag purification. Dialyzed His-tag purified nanobody fractions were diluted at 0.2 mg/ml concentration, which were directly used for immunostainings and western blots. To generate fluorescent-labeled nanobodies, Mix-n-Stain CF 555 Antibody Labeling Kit and Mix-n-Stain CF 647 Antibody Labeling Kit (Sigma-Aldrich, MX555S100 and MX647S100) were used according to the manufacturer's protocol.

Nanobodies equipped with an LPETGG motif and a His-tag at their C-terminus were functionalized with triglycine containing probes using Sortase 5 M as previously described (*Cheloha et al., 2019*). Briefly, LPETGG motif and His-tag at the C-terminus of nanobodies were replaced with fluorophore or biotin contained in triglycine probes by sortagging reaction. Ni-NTA resin (EMD Millipore, 70691–3) was used to remove unreacted His-tagged nanobodies and His-tagged Sortase 5 M enzyme, and the flow-through (sortase-based labeled nanobodies) was collected. The flow-through was applied to PD-10 column (GE Healthcare, GE17-0851-01) to remove excess triglycine probes.

## Immunostaining and imaging analysis

*Drosophila* fat body and midguts from adult females were fixed in 4% paraformaldehyde in phosphate-buffered saline (PBS) at room temperature for 1 hr, incubated for 1 hr in blocking buffer (5% normal donkey serum, 0.3% Triton X-100, 0.1% bovine serum albumin [BSA] in PBS), and stained with primary antibodies overnight at 4°C in PBST (0.3% Triton X-100, 0.1% BSA in PBS). S2*R*+ cells were fixed in 4% paraformaldehyde in PBS at room temperature for 30 min, incubated for 1 hr in blocking buffer, and stained with primary antibodies for 2–3 hr at room temperature in PBST. The primary antibodies and their dilutions used are: NbVHH05-ALFA (1:500), NbVHH05-HA (1:500), NbVHH05-hIgG (1:20, S2 cell culture media), NbVHH05-555 (1:500), Nb127D01-ALFA (1:500), Nb127D01-HA (1:500), Nb127D01-hIgG (1:20, S2 cell culture media), Nb127D01-647 (1:500), mouse anti-GFP (Invitrogen, A11120; 1:300), rat anti-HA (Sigma-Aldrich, 3F10; 1:1000), rabbit anti-Dilp2 (0.5 μg/ml) (*Park et al., 2014*). After primary antibody incubation, the fat body and midguts or S2*R*+ cells were washed three times with PBST, stained with 4',6-diamidino-2-phenylindole (DAPI) (1:2000 dilution) and Alexa Fluor-conjugated donkey-anti-mouse, donkey-anti-rabbit and mouse-anti-hIgG secondary antibodies (Molecular Probes, 1:1000), or NbALFA-Atto647 (NanoTag Biotechnologies, N1502-At647N-L; 1:500) Goat Anti-Alpaca IgG-647 (Jackson ImmunoResearch, 128-605-230; 1:500) in PBST at 22°C for 2 hr, washed three times with PBST, and mounted in Vectashield medium.

All images of the posterior midgut or S2*R*+ cells that are presented in this study are confocal images captured with a Nikon Ti2 Spinning Disk confocal microscope. Z-stacks of 5–20 images covering one layer of the epithelium from the apical to the basal side were obtained, adjusted, and assembled using NIH Fiji (ImageJ), and shown as a maximum projection.

## Western blots

Cultured cells were harvested 3 days after transfection. For fly, six to eight larval fat bodies or three female midguts per group were dissected in PBS, placed in 50 μl lysis buffer (Pierce, #87788) with 2× protease and phosphatase inhibitor cocktail (Pierce, #78440) and 2 mM trypsin inhibitor benzamidine

(Sigma-Aldrich, #434760), and homogenized using Kimble Kontes pellet pestles (Millipore, Z359947). Protein lysates were incubated in 2xSDS sample buffer (Thermo Scientific, #39001) containing 5% 2-mercaptoethanol at 100°C for 10 min, ran on a 4–20% polyacrylamide gel (Bio-Rad, #4561096), and transferred to an Immobilon-P polyvinylidene fluoride (PVDF) membrane (Millipore, IPVH00010). The membrane was blocked by 5% BSA or 5% skim milk in 1× Tris-buffered saline (TBS) containing 0.1% Tween-20 (TBST) at room temperature for 30 min. The following primary antibodies were used: anti-tubulin (Sigma, T5168, 1:10,000), rabbit anti-GFP (Molecular Probes, A-6455; 1:10,000) and anti-FLAG M1 (Sigma, F3040, 1:5000), rat anti-HA (Sigma-Aldrich, 3F10; 1:10,000), NbVHH05 (0.2 mg/ml, 1:5000, or 1:100~1:100,000 used in concentration gradient test), Nb127D01 (0.2 mg/ml, 1:5000, or 1:100~1:100,000 used in concentration gradient test), NbVHH05-hIgG (1:100, S2 cell culture media), Nb127D01-hIgG (1:100, S2 cell culture media) in blocking solution. After washing with TBST, signals were detected with enhanced chemiluminescence (ECL) reagents (Amersham, RPN2209; Pierce, #34095) or fluorescent secondary antibodies information. Western blot images were acquired by Bio-Rad ChemiDoc MP or X-ray film exposure.

## Immunoprecipitation

For immunoprecipitation using ALFA-tagged nanobodies (NbVHH05-ALFA and Nb127D01-ALFA), His-tag purified nanobodies from bacteria were incubated with ALFA Selector ST resin (Nanotag Biotechnologies, N1511) at room temperature for 1 hr. The resin was washed with Pierce IP lysis buffer (Thermo Scientific, 87787) (3×) and incubated with S2 cell culture media containing secreted GFP proteins with 3xVHH05-tag and 3x127D01 tag at 4°C for 1 hr. After washing in IP lysis buffer (4×), proteins were eluted in 2× sample buffer.

For immunoprecipitation using hIgG-formatted nanobody (Nb127D01-hIgG), Protein A magnetic beads (Bio-Rad, 1614013) were incubated with S2 cell culture media containing Nb127D01-hIgG at room temperature for 1 hr. After washing in IP lysis buffer (3×), beads were incubated with the conditioned media containing secreted GFP proteins 3x127D01 tag at 4°C for 1 hr. After washing in IP lysis buffer (4×), proteins were eluted in 2× sample buffer.

## ELISA

GFP fused at its C-terminus with a peptide corresponding to the extracellular portion of human CXCR2 (full-length, *Figure 1—figure supplement 1*), produced through sortagging, was immobilized on Nunc 96-well Maxisorp flat bottom plates (100 ng/well). After immobilization, wells were blocked using a solution of BSA in PBS (5% w/v). Nb127D01 conjugated with biotin (20 nM) was mixed with peptides corresponding to full-length CXCR2 extracellular domain or two fragments at varying concentrations. These solutions were then added to plates with immobilized GFP-CXCR2 and incubated for 1 hr. These solutions were discarded from the plates, the plates were washed with PBS containing 0.05% Tween-20, and the amount of Nb127D01-biotin bound to the plate was quantified through the addition of streptavidin-HRP (Thermo Fisher, N100), washing, and the addition of tetramethylbenzidine-containing solution (Thermo Fisher, N301).

## Trehalose assay

Whole-body trehalose levels were measured from five to six groups (each group has four female flies). Fly samples were homogenized with 300 µl TBST buffer (5 mM Tris-HCl [pH 6.6], 137 mM NaCl, 2.7 mM KCl, 0.1% TritonX-100), heated at 75°C for 10 min, and centrifuged at 3000 $g$ for 1 min. Ten µl of supernatant was added to 100 µl glucose assay reagent (Megazyme; K-GLUC) with or without trehalose (1:500; Megazyme; E-TREH) at 37°C for 30 min. The absorbance at 340 nm was measured on a Molecular Devices SpectraMax Paradigm plate reader. The trehalose concentration in the sample was determined by subtracting the glucose concentration from the total sugar concentration.

## Acknowledgements

We thank the assistance provided by the Microscopy Resources on the North Quad (MicRoN) core at Harvard Medical School, Christians Villalta for help with the generation of transgenic flies and Hong-Wen Tang for help with the Western blots. We thank Seung K Kim for providing the anti-Dilp2 antibody. This work was supported by NIH NIGMS P41 GM132087. AK is supported by Postdoctoral Fellowship Program (Nurturing Next-generation Researchers) through the National Research

Foundation of Korea (NRF) funded by the Ministry of Education (2021R1A6A3A14039622). JSSL is supported by a Croucher fellowship for Postdoctoral Research from the Croucher Foundation. NP is an investigator of Howard Hughes Medical Institute.

## Additional information

### Funding

| Funder | Grant reference number | Author |
| --- | --- | --- |
| National Institute of General Medical Sciences | GM132087 | Norbert Perrimon |
| National Research Foundation of Korea | 2021R1A6A3A14039622 | Ah-Ram Kim |
| Croucher Foundation | | Joshua Shing Shun Li |
| Howard Hughes Medical Institute | | Norbert Perrimon |

The funders had no role in study design, data collection and interpretation, or the decision to submit the work for publication.

### Author contributions

Jun Xu, Ah-Ram Kim, Conceptualization, Data curation, Formal analysis, Investigation, Methodology, Resources, Software, Validation, Visualization, Writing – original draft, Writing – review and editing; Ross W Cheloha, Conceptualization, Formal analysis, Investigation, Methodology, Resources, Validation, Visualization, Writing – review and editing; Fabian A Fischer, Formal analysis, Investigation, Visualization; Joshua Shing Shun Li, Formal analysis, Investigation, Writing – review and editing; Yuan Feng, Emily Stoneburner, Formal analysis, Investigation, Methodology, Validation; Richard Binari, Formal analysis, Investigation, Resources; Stephanie E Mohr, Conceptualization, Data curation, Project administration, Resources, Writing – review and editing; Jonathan Zirin, Data curation, Methodology, Project administration, Resources, Validation, Writing – review and editing; Hidde L Ploegh, Conceptualization, Data curation, Funding acquisition, Methodology, Resources, Software, Supervision, Validation, Writing – review and editing; Norbert Perrimon, Conceptualization, Data curation, Funding acquisition, Methodology, Project administration, Resources, Supervision, Validation, Writing – review and editing

### Author ORCIDs

Jun Xu http://orcid.org/0000-0002-7963-0253
Ah-Ram Kim http://orcid.org/0000-0001-9597-6759
Stephanie E Mohr http://orcid.org/0000-0001-9639-7708
Norbert Perrimon http://orcid.org/0000-0001-7542-472X

### Decision letter and Author response

Decision letter https://doi.org/10.7554/eLife.74326.sa1
Author response https://doi.org/10.7554/eLife.74326.sa2

## Additional files

### Supplementary files

• Supplementary file 1. Primers used in this study.

• Transparent reporting form

### Data availability

All data generated or analysed during this study are included in the manuscript and supporting file.

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
