## [Decision Letter]

**Decision letter after peer review:**

Thank you for submitting your article "Protein visualization and manipulation in *Drosophila* through the use of epitope tags recognized by nanobodies" for consideration by *eLife*. Your article has been reviewed by 2 peer reviewers, and the evaluation has been overseen by a Reviewing Editor and K VijayRaghavan as the Senior Editor. The following individuals involved in review of your submission have agreed to reveal their identity: Gerald M Rubin (Reviewer #1); Fillip Port (Reviewer #2).

Essential revisions:

The revisions asked for concern matters of presentation and can hopefully be done speedily.

*Reviewer #1:*

This Tools and Resources paper makes a very significant contribution to the *Drosophila* toolbox by addressing the lack of multiple, well-characterized epitope/nanobody pairs for use in vivo and in vitro in *Drosophila*. Two new epitope/nanobody pairs and their characteristics are described in detail. These reagents, together with the previously described ALFA tag, fill the need for multiple independent epitope/nanobody pairs in experiments involving co-localization and other applications. The experiments are well-done, carefully documented and support the conclusions drawn. Methods for direct labelling of the nanobodies with fluorescent dyes are described, an important method for quantitative work. CRISPR-mediated knock-in of the tags into endogenous genes is also demonstrated. The nanobodies also work well in protein immunoblots and in biochemical purifications.

Page 1, line 29: the ")" after VHH05 should be a ",".

Page 3, line 17-18: "NbVHH05 (Ling et al., 2019) and Nb127D01". Bradley et al., 2015 reference should be inserted after Nb127D01.

In Figure 1: "Mitochondrial" should be replaced with "Mitochondria" in panels G and J. Why are the nuclei blue in color in Panel I? If DAPI stained, please say so and indicate which panels also included DAPI staining. In panel E, add to legend that the right most panel is GFP, the center panel is mCherry and the rightmost is a merged image.

In Figure 1, figure supplement 2: State size of scale bars in images.

Page 8, line 20: fluorophores (NbVHH05-555 and Nb127D01-647). Specify the identity of the fluorescent dyes used.

Page 8, line 28: It would help the reader if a one sentence explanation of sortase labeling was added here. Most readers will be unfamiliar with this method.

Page 8, line 30: Page 8, line 30: It would be more accurate to say "highly" than "absolutely" here.

Figure 2. Merged images appear to also have DAPI (or other nuclear) stain, but this is not mentioned in the legend.

Figure 2—figure supplement 1. Panels are mislabeled (A is really C). Define "FT" as flow thru.

Figure 2—figure supplement 3. "Con-cell" is a little cryptic. "Mock transfected" would be better.

Page 24 line 25: Please explain "KI method". I think this is the first time the "KI" abbreviation is used.

Page 28, line 30: It was unclear to this reader what "CB" referred to. I had to go back thru the paper to find the definition. It would be better to spell it this location (the first sentence of the discussion).

*Reviewer #2:*

This work characterizes two nanobodies and the short peptide sequences they recognize for protein detection and manipulation in vitro and in vivo. Nanobodies are natural, short protein binders that can be genetically encoded and expressed in living cells, enabling many applications not feasible with conventional antibodies. Nanobodies that recognize short epitope tags have recently been described, but their number is currently limited. This work expands the repertoire of nanobody/nanotag pairs and demonstrates their usefulness in a large range of applications.

This paper is well written, the experiments are thoroughly designed and the data unambiguously support the conclusion that the two described nanobody/nanotag pairs are useful tools for cell and molecular biology. A particular strength of this study is the breadth of assays in which the authors test the described tools, ranging from in vitro assays such as Western blotting and immunoprecipitation to in vivo assays such as protein detection with fluorophore-tagged nanobodies or protein relocalization. Furthermore, the authors test a range of secondary reagents to detect unlabeled nanobodies and showcase how CRISPR genome engineering can be used to insert sequences encoding the nanotags into endogenous loci.

Going forward it will be important to better understand the specificity of the two nanobodies. Perfect matches of the corresponding nanotag sequences are not encoded in the *Drosophila* genome, as well as a range of other relevant genomes, but nanobodies might bind to endogenous proteins with imperfect sequence homology. The finding that nanobodies can be expressed in vivo without obvious fitness defects suggests that off-target binding is not prevalent, but does not exclude binding to specific targets. In the future it will therefore be important to more directly characterize potential endogenous binding partners of the two nanobodies. This could be done by immunoprecipitation of cell lysates followed by mass-spectrometry or by fusing nanobodies to a biotin ligase and expressing it in cells, followed by identification of biotinylated proteins.

Overall, this is a well-designed study that presents an exciting pair of tools that will be of broad interest to anyone interested in detecting and manipulating proteins in vitro, in cells or in living organisms.

Page 1; Line 29: (remove from VHH05)

The graphics that are provided alongside the figures are helpful to quickly grasp the experimental setup. However, they could be yet more intuitive if the used shapes adhered to common praxis. In particular, antibodies (e.g. Figure 2 C, D and others) should be depicted in their prototypical Y shape, rather than rectangular shapes.

Page 20, lines 23 – 38: The authors generate several constructs with the aim to check "whether NanoTagged proteins can be detected in vivo". However, in the end only in vivo detection of Upd2 (and Dilp2 in a later figure) is reported. What happened to the constructs encoding tagged Akh, Dilp8, Pvf1 and Reptor-BP? Were they not tested in vivo?

Page 24, lines 2 – 6: Another common feature are the encoded NanoTags.

Page 29, line 19: The observation of protein aggregates should also be mentioned in the Results section when nanobody-GFP fusions are initially tested in S2R+ cells.

---

## [Author Response]

Essential revisions:The revisions asked for concern matters of presentation and can hopefully be done speedily.Reviewer #1 (Recommendations for the authors):Page 1, line 29: the ")" after VHH05 should be a ",".

We fixed this mistake.

Page 3, line 17-18: "NbVHH05 (Ling et al., 2019) and Nb127D01". Bradley et al., 2015 reference should be inserted after Nb127D01.

We added the reference as suggested.

In Figure 1: "Mitochondrial" should be replaced with "Mitochondria" in panels G and J. Why are the nuclei blue in color in Panel I? If DAPI stained, please say so and indicate which panels also included DAPI staining. In panel E, add to legend that the right most panel is GFP, the center panel is mCherry and the rightmost is a merged image.

Thank you for the suggestion. We replaced "Mitochondrial" with "Mitochondria" in panels G and J. We also added in the legend:

"the right most panel is GFP, the center panel is mCherry, and the rightmost is the merged image. DAPI staining shows the nuclei ".

In Figure 1, figure supplement 2: State size of scale bars in images.

We added "Scale bars: 25 μm" in the legend.

Page 8, line 20: fluorophores (NbVHH05-555 and Nb127D01-647). Specify the identity of the fluorescent dyes used.

Thank you for the comment. We replaced NbVHH05-555 and Nb127D01-647 with “NbVHH05-CF555 and Nb127D01-CF647” as suggested. In addition, we edited the other NbVHH05-555 prepared by sortase reaction as “NbVHH05-AF555”.

Page 8, line 28: It would help the reader if a one sentence explanation of sortase labeling was added here. Most readers will be unfamiliar with this method.

Thank you for the suggestion. We have added an explanation of the sortase labeling method. "In addition, we used a fluorescent NbVHH05-AF555 prepared by site-specific Sortase labeling (Figure 2—figure supplement 2A3). For C-terminal Sortase-mediated labeling, the Sortase recognition motif (LPETG) is added to the C-terminus of the nanobody. Next, catalyzed by the Sortase, a fluorophore or biotin is added, using modified oligoglycine peptides such as GGG-fluorophore or GGG-biotin, to the nanobody-LPETG (*Cheloha et al., 2019*)."

Page 8, line 30: Page 8, line 30: It would be more accurate to say "highly" than "absolutely" here.

We replaced "absolutely" with "highly”.

Figure 2. Merged images appear to also have DAPI (or other nuclear) stain, but this is not mentioned in the legend.

We added "DAPI staining shows the nuclei " into the legend.

Figure 2—figure supplement 1. Panels are mislabeled (A is really C). Define "FT" as flow thru.

We revised the mislabeled panel and defined "FT" as Flow-Through in the legend.

Figure 2—figure supplement 3. "Con-cell" is a little cryptic. "Mock transfected" would be better.

We replaced the "Con-cell" with "Mock transfected".

Page 24 line 25: Please explain "KI method". I think this is the first time the "KI" abbreviation is used.

We replaced the "KI method" with "knock-in (KI) method".

Page 28, line 30: It was unclear to this reader what "CB" referred to. I had to go back thru the paper to find the definition. It would be better to spell it this location (the first sentence of the discussion).

We replaced the "CB" with "chromobodies (CBs)".

Reviewer #2:Page 1; (Line 29: remove from VHH05)

We removed ")" as suggested.

The graphics that are provided alongside the figures are helpful to quickly grasp the experimental setup. However, they could be yet more intuitive if the used shapes adhered to common praxis. In particular, antibodies (e.g. Figure 2 C, D and others) should be depicted in their prototypical Y shape, rather than rectangular shapes.

Thank you for the suggestion. We changed the antibodies in Figure 2 C, D with Y shape.

Page 20, lines 23 – 38: The authors generate several constructs with the aim to check "whether NanoTagged proteins can be detected in vivo". However, in the end only in vivo detection of Upd2 (and Dilp2 in a later figure) is reported. What happened to the constructs encoding tagged Akh, Dilp8, Pvf1 and Reptor-BP? Were they not tested in vivo?

We added “in vitro” in the sentence to clarify this point (“Next, we checked whether NanoTagged proteins can be detected in vitro and in vivo using purified nanobodies.”).

Originally, we chose Upd2 as an example for in vivo detection. In the meantime, we succeeded at generating transgenic REPTOR-bp-C flies and the staining results showed that NanoTagged Reptor-BP can be detected, which we added as new results in Figure 5—figure supplement 2.

Page 24, lines 2 – 6: Another common feature are the encoded NanoTags.

We modified the sentence (“Each vector contains five common features: the encoded NanoTags for N- or C-terminal tagging, …”).

Page 29, line 19: The observation of protein aggregates should also be mentioned in the Results section when nanobody-GFP fusions are initially tested in S2R+ cells.

As suggested, we now mention in the Results section: "When either NbVHH05-GFP or Nb127D01-GFP was expressed in cells, we observed a GFP signal in the nucleus and the cytoplasm, although some S2R+ cells transfected with Nb127D01-GFP contained aggregates (Figure 1—figure supplement 2C,D)."